# Atmospheric processes affecting the separation of volcanic ash and SO$_2$ in volcanic eruptions:

# Inferences from the May 2011 eruption

Fred Prata[1], Mark Woodhouse[2], Herbert E. Huppert[3], Andrew Prata[4], Thor Thordarson[5], and Simon Carn[6]

[1]Visiting scientist, Department of Atmospheric, Oceanic and Planetary Physics, Clarendon Laboratory, University of Oxford, UK

[2]School of Mathematics, University of Bristol, Clifton, Bristol, UK

[3]Institute of Theoretical Geophysics, Department of Applied Mathematics and Theoretical Physics, University of Cambridge, UK

[4]Department of Meteorology, University of Reading, Earley Gate, Reading, UK

[5]Faculty of Earth Sciences, University of Iceland, Reykjavik, Iceland

[6]Department of Geological and Mining Engineering and Sciences, Michigan Technological University, Houghton, MI, USA

*Correspondence to:* Fred Prata (fred_prata@hotmail.com)

**Abstract.** The separation of volcanic ash and sulphur dioxide (SO$_2$) gas is sometimes observed during volcanic eruptions. The exact conditions under which separation occurs are not fully understood but the phenomenon is of importance because of the effects volcanic emissions have on aviation, on the environment and to the earth's radiation balance. The eruption of , a subglacial volcano under the Vatnajökull glacier in Iceland during 21–28 May 2011 produced one of the most spectacular examples of ash and SO$_2$ separation that led to errors in the forecasting of ash in the atmosphere over northern Europe. Satellite data from several sources coupled with meteorological wind data and photographic evidence suggest that the eruption column was unable to sustain itself, resulting in a large deposition of ash which left a low level ash-rich atmospheric plume

moving southwards and then eastwards towards the southern Scandanavian coast, and a high level predominantly $SO_2$ plume travelling northwards and then spreading eastwards and westwards. Here we provide observational and modelling perspectives on the separation of ash and $SO_2$ and present quantitative estimates of the masses of ash and $SO_2$ erupted, the directions of transport, and the likely impacts. We hypothesise that a partial column collapse or "sloughing" fed with ash from pyroclastic density currents (PDCs) occurred during the early stage of the eruption leading to an ash-laden gravity intrusion that was swept southwards, separated from the main column. Our model suggests that water-mediated aggregation caused enhanced ash removal because of the plentiful supply of source water from melted glacial ice and from entrained atmospheric water. The analysis also suggests that ash and $SO_2$ should be treated with separate source terms, leading to improvements in forecasting the movement of both types of emissions.

## 1 Introduction

Vigorous volcanic eruptions emit copious amounts of gases and particles into the atmosphere where they are transported by the winds, potentially in all directions. They can be transported rapidly zonally as in the case of the eruption of Puyehue-Córdon Caulle, southern Chile during June 2011 where ash and $SO_2$ travelled together circling the southern hemisphere at latitudes south of $30\,^\circ$S. They can be transported vertically by air circulations as in the case of Nabro, Eritrea also in June 2011 where the monsoon circulation may have played a part in lifting $SO_2$ gas into the stratosphere (Bourassa et al., 2012), although Fromm et al. (2014) provides a convincing case for direct gas injection. Prevailing atmospheric winds can play a pivotal role in the transport of ash and $SO_2$ as in the case of the April and May 2010 eruptions of Eyjafjallajökull, Iceland, during which large amounts of ash were transported zonally and meridionally over continental Europe leading to major disruptions of air traffic. The direction of transport is determined by the strength and direction of the zonal and meridional wind fields and these vary with height. Vertical wind shear varies with location and time and is commonplace.

$SO_2$ gas and ash particles represent two major components of vigorous volcanic activity and these may be emitted together or individually and this mix can and does vary with time, due largely to the character of the volcanic activity and the geological setting of the volcano. Since there is no guarantee that ash and $SO_2$ will be erupted at the same time, nor that they will remain collocated in space and time, there is a good reason to investigate the conditions under which these emissions remain together and conditions under which they separate. Magma composition is also a factor because lower viscosity magmas fragment into coarser particles that separate more easily from gas. Separation has been observed during the eruptions of Okmok and Kasatochi (Prata et al., 2010), this eruption (Sigmarsson et al., 2013; Moxnes et al., 2014) and during the Eyjafjallajökull eruption (Thomas and Prata, 2011), although collocated transport is also often observed. Holasek et al. (1996) investigated ash and gas separation through analogue laboratory experiments. They found that gas and ash separation occurred through buoyancy effects with particle sedimentation leaving higher gas concentrations above. Separation can also occur when ash and $SO_2$ are emitted in separate explosive pulses where either the energy of the eruption has changed, emplacing the materials at different heights, or the atmospheric conditions have changed in the intervening period. These processes are complex and difficult to predict for individual events.

Here we study the remarkable separation of SO$_2$ and ash during the 21–28 May, 2011 eruption of , using mostly satellite data but also ground based remote sensing measurements and model simulations. The separation was the greatest possible. SO$_2$ reached high altitudes (>10 km), travelled northwards and then spread eastwards and westwards, while the ash remained at low altitudes (<4 km) travelled southwards, before spreading eastwards, eventually reaching the western coast of Norway. The separation led to a poor forecast of the ash hazard to aviation and we suggest how such errors can be avoided in the future.

The injection of SO$_2$ into the stratosphere, and its subsequent conversion to sulphate aerosol is important for understanding the radiative impact of volcanic eruptions (Robock, 2000). If the SO$_2$ emission remains largely in the troposphere and is of short duration (less than a few days) its potential impact on radiative forcing is less, because the residence time of the resulting sulphate aerosol is shorter. Stratospheric sulphate aerosols on the other hand can potentially alter the radiative balance. It is shown here that the SO$_2$ did indeed penetrate into the stratosphere. The satellite data are used to estimate the total SO$_2$ injected into the stratosphere, and the total mass of very fine ash injected into the lower troposphere. Here the term 'very fine' is used to describe ash with an effective radius < 16 $\mu$m, which represents the largest grain size that IR sensors and retrieval algorithms can quantifying with any certainty.

Our paper stresses the importance of a multidisciplinary approach to the study of volcanic eruptions on the atmosphere: volcanolgical insights, space-based observations, dispersion modelling and fluid dynamics are all required to develop understanding of the dominant processes involved. The paper is organised as follows. The chronology of the eruption and important events are described, followed by a short section on the transport and the tools used to determine it. Next the satellite data are introduced and estimates of the ash mass loading, the SO$_2$ amount, cloud-top temperature and height are provided. The phenomenon of particle-gas separation is discussed and observational evidence is presented for the eruption. An uncoupled plume model, in which plume dynamics and plume microphysics are examined separately is used to provide insights into the most significant processes relevant to particle-gas separation. Most of the satellite observations and some modelling results are included as Supplementary Material. The main inferences from the study are presented in a concluding discussion section and an Appendix provides a mathematical description of the plume model employed.

## 2   Chronology of the eruption

is a subglacial volcano situated under the Vatnajökull glacier in south-eastern Iceland. Like many Icelandic volcanoes it has a long record of eruptive activity (Thordarson and Larsen, 2007) with the last notable event prior to the May 2011 activity occurring between 1–6 November, 2004. On the afternoon of 21 May 2011 at around 17:30 UTC, seismicity and thermal anomaly measurements at indicated that an eruption was likely, and at 19:00 UTC the eruption penetrated the subglacial caldera. The first signs from satellite observations of a plume entering the atmosphere were recoded by the Spin-Enhanced Visible and Infra-Red Instrument (SEVIRI) on board the geostationary Meteosat Second Generation (MSG)-2 satellite at 19:15 UTC on 21 May. The weather conditions at the start of the eruption were good and photographs of the plume (see Fig. 1 and the Supplementary material) as it emerged out of the glacier at ∼19:10 UTC, clearly showed a steam-rich plume that later developed into an ash laden plume reaching several kilometres into the atmosphere. As evening fell, visibility worsened and

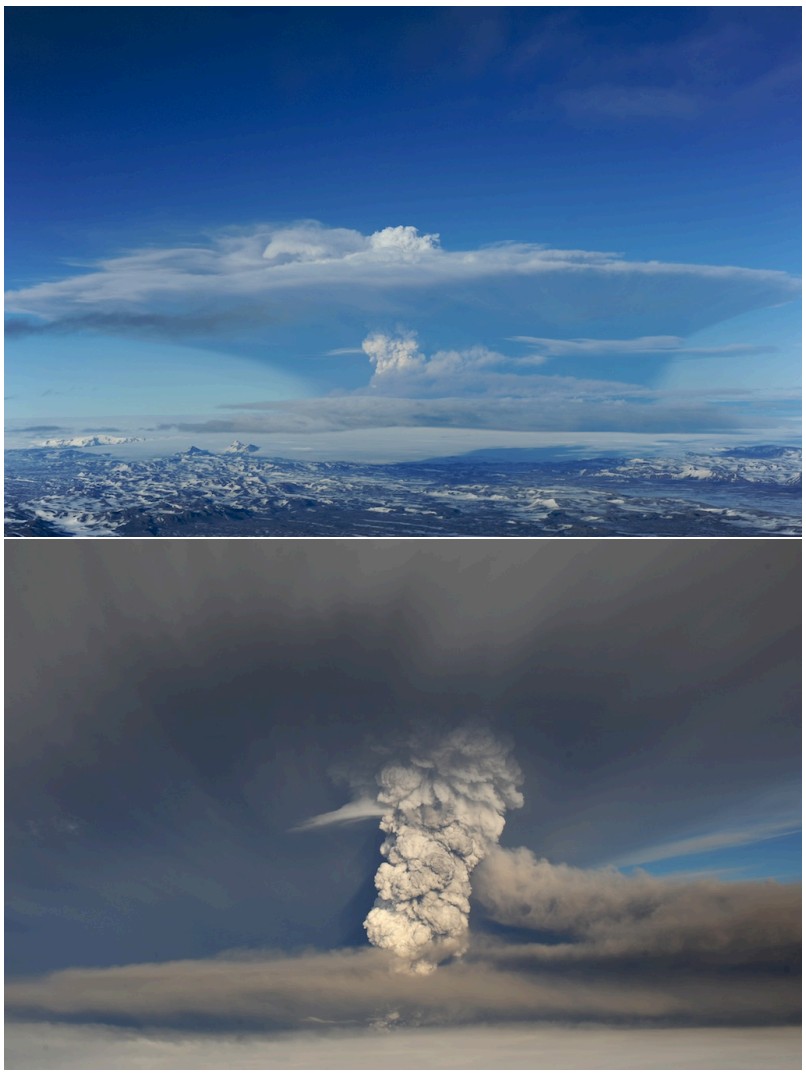

**Figure 1.** The changing appearance of the eruption column in photographs taken at the start of the eruption in clear skies at 20:10 UTC (20:10 local time) where the whitish appearance of the column suggests condensed water vapour (top), and later (21:05 UTC) in a dark (ash-rich), cloud-laden atmosphere (bottom). There is a clear indication (lower photograph) of ash in the lower parts of the column. Later photographs show mammatus clouds forming in the eruption cloud suggesting ice nucleation and this may have contributed to a more rapid loss of ash particles from the cloud. Photographs courtesy of Ólafur Sigurjónsson. See also Supplementary Photographs.

cloud also moved in from the north making visual identification of the plume difficult. Early reports and some later analysis suggested that the plume had reached perhaps 15–19 km (see also the Supplementary photographs). Ashfall was evident all around  and was reported from the Reykjavik area in the SW to Tröllaskagi Peninsula in the north. Figure 2 shows a map of the region indicating the location of , some of the towns and the airport.

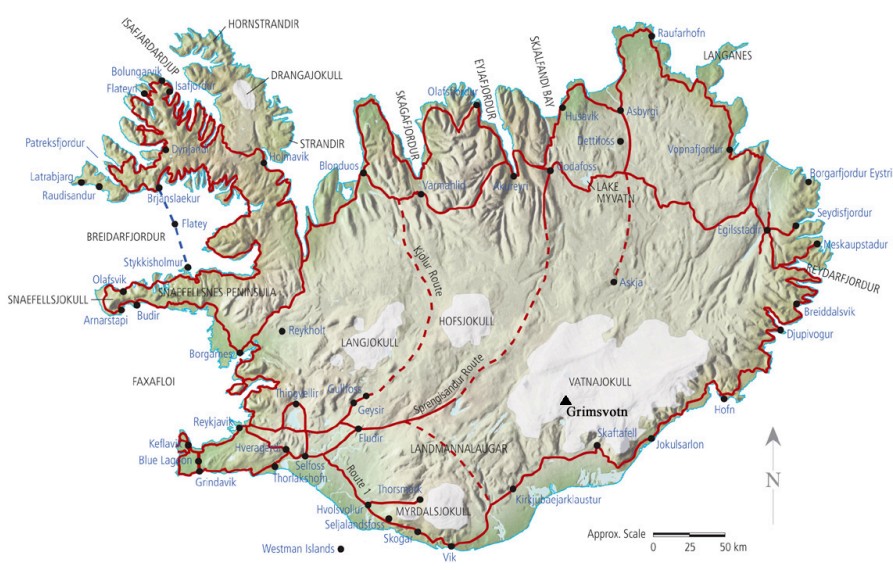

**Figure 2.** Map of Iceland showing towns, the road network and the location of .

According to the status reports (http://earthice.hi.is/grimsvotn_eruption_2011) issued by the Icelandic Meteorological Office (IMO) and the Institute for Earth Sciences (IES) in Iceland, the column reached greatest heights during 21–22 May which were estimated to be between 15–19 km. On 24 May the column height was between 5–7 km and on 25 May it was below 5 km. Subsequently the columns dropped to 10 km and then by 26 May did not exceed 8 km and mostly remained below cloud level, with a white steam plume observed to reach 2 km. The  eruption is believed to be the largest in Iceland since the eruption of Katla in 1918, and erupted a tephra volume of $\sim$0.7 km$^3$ (Gudmundsson et al., 2012a) (or $\sim$0.15 km$^3$ dense rock-equivalent, based on our average vesicularity measurements of $\sim$78%) in just over 2 days compared with the April-May 2010 Eyjafjalla-jökull which erupted $\sim$0.27 km$^3$ over a 39 day period (Gudmundsson et al., 2012b). Sigmarsson et al. (2013) estimate a total of $\sim$0.2 Tg(S) was released to the atmosphere. Stevenson et al. (2012) demonstrated that ash from Eyjafjallajökull reached many parts of the United Kingdom, noting that even quite large ($>$50 $\mu$m radii) particles were collected on the ground. The much larger  eruption was expected to send more ash towards Europe than the April-May 2010 Eyjafjallajökull event that caused Europe-wide aviation disruption.

## 15  3  Transport

Atmospheric dispersion models are used to simulate the transport of volcanic ash in the atmosphere, so-called Volcanic Ash Transport and Dispersion (VATD) models (Draxler and Rolph, 2003, e.g. HYSPLIT). These models have been quite successful for both small (e.g Eyjafjallajökull) and mid-size (e.g. Puyehue-Córdon Caulle) eruptions. They depend for their accuracy on the precise details of the eruption parameters, and in particular on being able to specify the mass eruption rate (MER), its

vertical structure and its temporal variation. If these are incorrect then the forecast dispersion is also incorrect. Many VATD models rely on an estimation of the MER obtained from a parameterised relationship between the MER and the height of the eruption column. The most commonly used parametrisation has the MER proportional to the fourth power of the height (Sparks et al., 1997b; Mastin et al., 2009) under the assumption of dry standard atmosphere conditions, without the inclusion of ice/water at the vent, which is likely a large factor in the case of the  eruption. There are two significant consequences

for forecasting the transport if the height is in error. First, for an atmosphere with significant wind shear where the height is incorrectly specified, the transport will be incorrect. Second, as the MER depends strongly on height, the estimated amount of ash may be significantly under- or over-estimated, if the height is under- or over-estimated. More sophisticated models of the MER are available (Mastin et al., 2009), and recent work by Woodhouse et al. (2013) among others has shown a dependence of the MER on wind shear for bent-over plumes Degruyter and Bonadonna (2012). These more detailed treatments also require

more detailed observations of various parameters to specify the MER.

HYSPLIT ash dispersion runs using GDAS wind fields were used to test the sensitivity of the transport to the height of the eruption column during the  eruption. Inspection of the photographs (see Supplementary Photographs, especially Figure S2) shows that the initial plume is just steam (water vapour) and condensed water vapour, but by 19:30 UTC the column appears to be ash-laden and fully developed by 20:00 UTC. At 20:30 UTC the plume at the top of the column appears to lighten suggesting

that the transition from ash-rich to gas-rich upper plume was complete. Estimating the heights from the photographs is difficult but if it is assumed that the maximum height reached is ∼19 km, then by 19:10 UTC it is ∼8 km and by 19:25 UTC it is ∼16 km. For a column rising to ∼9 km or higher beginning on 21 May at 19:15 UTC the transport of ash is mostly northwards and then spreads eastwards and westwards. Conversely, for ash emitted to a height of ∼3 km on 22 May at 14:00 UTC the transport is mostly southwards and then eastwards towards Scotland and on to the southern part of Scandanavia. These two scenarios

are motivated by satellite observations of volcanic emissions taken by polar and geostationary instruments (see next section) and are therefore indicative of what actually happened. During the event, the London Volcanic Ash Advisory Centre (VAAC), among others, forecast ash emissions towards the north and south, in broad agreement with the HYSPLIT simulations. As we shall see, the forecast of significant ash emissions to the north was incorrect because these emissions were almost entirely $SO_2$ gas, and the forecast overestimated the amount of ash transported to the south and then eastwards, probably due to an

incorrectly specified vertical distribution of ash at the source.

A more complex model written specifically for volcanic ash eruptions, FALL3D (Folch et al., 2009) was also used to study the transport. The model is an Eulerian dispersion model which includes several different parametrisations that can used to specify the source term. It is driven by input atmospheric wind fields and has been used in many studies of atmospheric

transport of ash, particularly to investigate ash fall. We used FALL3D initialised with NCEP wind fields at 6-hourly intervals starting at 19:00 UTC on 21 May 2011. The source term was specified using one of the pre-set options, in this case a point source and the plume options were used. A forward run was performed on a grid of 0.2 x 0.2 degree resolution, with a vertical scale of ∼0.1 km for a total duration of up to 96 hours. Here we are not concerned with testing the model's propensity to accurately simulate long-range ash transport, but rather to investigate the hypothesis that the source of ash was from a column

collapse, with the ash injection treated as an impulse. Some of the results of the simulations are shown in the Supplementary Material section (Figure S1), at 6-hourly intervals starting at 13:00 UTC on 22 May. Here we summarise the main findings, noting that FALL3D is used for ash forecasting and so the results are 'typical' of what is found from using state-of-the-science VATD models.

The short duration of the source of ash results in a small plume or cloud of ash dispersing to the SSW and then turning

southwards (Fig. S1) and later towards the east (not shown). The ash eventually is transported over the northern part of the United Kingdom, somewhat south of the path observed in SEVIRI, and then onto the southern part of Scandanavia, where ground-based particle measurement stations recorded elevated levels of PM10 (Prata and Prata, 2012). The FALL3D simulations also show a small amount of ash transported towards the north-west at higher levels, collocated with satellite observations of $SO_2$. This ash cloud, also observed in ash retrievals from SEVIRI, the Infrared Atmospheric Spectrometer Interferometer (IASI) and Moderate Resolution Spectroradiometer (MODIS), is short-lived and has dissipated by 20:00 UTC on 22 May.[1] The

initial amount of erupted very fine ash required to generate an ash cloud consistent with the satellite observations cannot be modelled accurately using the fourth power law, as this produces almost 100 times too much total mass so that the mass fraction of very fine ash must then be altered. A scenario more consistent with the satellite estimates is that after the tephra-laden plume reached full development around 19:30–20:00 UTC, rising to height of ∼19 km, its carrying capacity dropped dramatically

over the next hour or so, resulting in the erupted tephra separating from the gas phase of the plume and spreading outwards as a gravitational current at much lower altitude, i.e. around ∼8 (6–10) km a.s.l. This change may have been induced by widening of the erupting vent, inducing partial column collapse and formation of pyroclastic density currents (PDCs). Combined effects of ash aggregation and a co-PDC plume would have enriched ash content of the outward moving lower-level plume. FALL3D is not currently configured to simulate these kinds of source terms (areal), so the source term was scaled to match the satellite

retrievals, assuming a maximum height of 6 km and with a suitable vertical distribution of ash. The MER scales as $H^4$, where $H$ is the column height, so the MER is reduced by a factor of ∼100 if the column height is reduced from 19 km to 6 km.

Stevenson et al. (2013) studied ash deposits in Scotland and northern England using pollen traps, tape-on-paper measurements, rainwater samples and air quality measurements. The majority of ash deposition was found in Scotland in agreement with the observations presented here. However, they find larger median grain sizes of 19–23 $\mu$m and maxima of 80 $\mu$m signif-

icantly higher than the effective radii found in the atmosphere from the satellite measurements. Since Stevenson et al. (2013) sample ash on the ground and have a bias which precludes making measurements of gran sizes <10 $\mu$m and the infrared retrievals are less sensitive to particles with effective radii > 10 $\mu$m, the two measurements strategies are largely incompatible. Very fine ash was also detected by the infrared sensors over Greenland during the early phase of the eruption on 22 May

---

[1]Here it may be assumed that the ash is still in the atmosphere but of a concentration too low to be detected by current satellite infrared measurements.

(Prata and Rose, 2015, see Figure 52.11). This ash signal quickly dissipates and the ash was high in the atmosphere (above the tropopause), and collocated with the $SO_2$. There is no evidence from the satellite measurements that any of this ash reached the UK or Europe. Any ash arriving in Europe from this part of the plume would have had to descend from the stable stratosphere and having travelled for 4–5 days would be of low concentration and have a very small effective radius.

### 3.1 Separation of the dispersing volcanic cloud

The observational evidence for significant separation of ash and $SO_2$ is unequivocal, but was this entirely due to wind shear? Certainly at some stages during the eruption the column reached at least 16 km and wind data show that wind shear was present, which suggests the emissions would disperse in different directions. During the period between the early morning of 22 May and late afternoon of 23 May, satellite, radar and photographic evidence shows that the column was changing height and at times extending to less than 10 km. AIRS near real-time brightness temperatures can be used to retrieve upper level

$SO_2$ in the atmosphere. A series of six images of this product are shown in Figure S2. These products are based on brightness temperatures at specific wavenumbers that correspond to $SO_2$ absorption locations and a radiative transfer model is used to fit the spectral features to the $SO_2$ column amount (Prata and Bernardo, 2007). A more simplified product is also used, indicating $SO_2$ as negative temperature differences, where $\Delta T < -6K$. Positive differences found in these images are due to other causes and strongly positive values are generally unusual. The causes of positive differences are mostly associated with water vapour

and thermal contrast effects. There is no other literature that we are aware of discussing positive differences in this product. Figure 3 shows three of these brightness temperature difference products.

The $SO_2$ feature is clearly evident in these images, but there is also a positive difference coincident with the location of the  vent. We speculate that the positive temperature differences correspond to the location of the ash-rich eruption column, slightly displaced from the upper level dispersing $SO_2$. Interestingly, by 04:17 UTC on 23 May 2011 this positive anomaly

has disappeared. The brightness temperature difference is determined from two channels at 1361.44 $cm^{-1}$ and 1433.06 $cm^{-1}$ situated inside and outside of the strong $\nu_3$ $SO_2$ absorption band (Prata and Bernardo, 2007). The reason for the positive difference over the column is unclear, but we suggest that because these two channels are sounding the atmosphere with peak contributions at different heights, a positive difference will be observed when the top of the column is below the upper level sounding channel (1433.06 $cm^{-1}$) peak, and above or near to the weighting function peak of the lower level channel (1361.44

$cm^{-1}$). This interesting observation suggests that it may be possible to utilise the AIRS spectral information to determine ash column heights for opaque columns. The changes in height of the ash column and vertical emplacement of ash is a significant process that affects the subsequent direction of transport of volcanic ash.

## 4    Satellite data analyses

Satellite instruments have been used extensively to monitor volcanic emissions (Prata, 2009). Both ash and $SO_2$ gas can be

quantified using measurements made in the infrared (e.g. Clarisse et al., 2010; Prata and Prata, 2012; Carn et al., 2005), and in the ultraviolet, (e.g. Carn et al., 2016), while UV-visible reflected light can be used to identify volcanic aerosols (aerosol optical

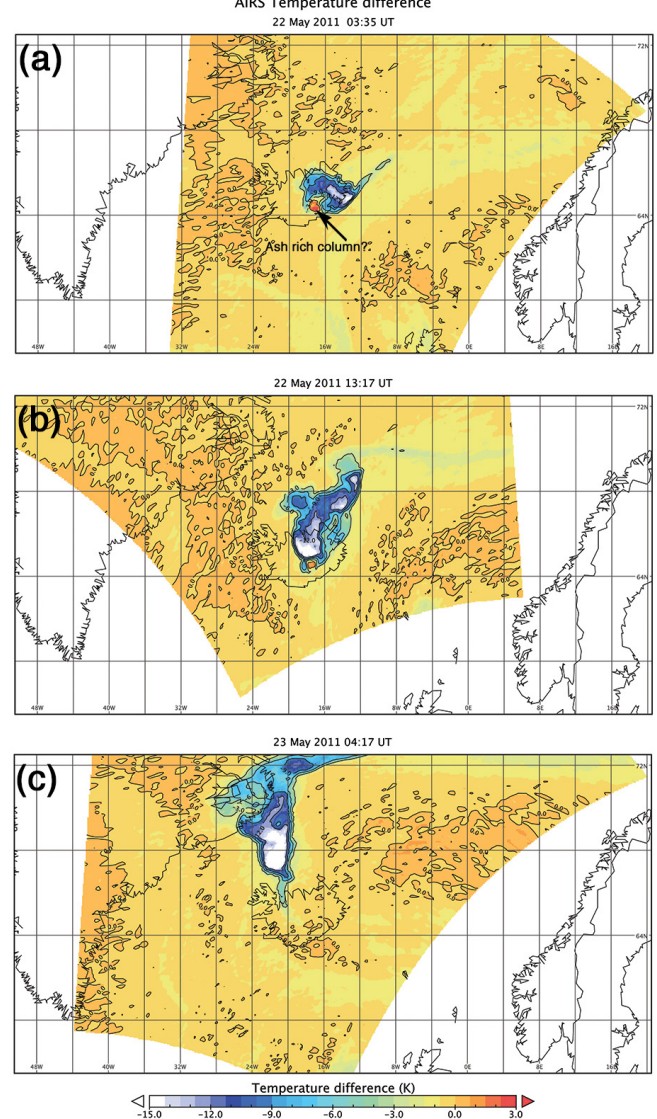

**Figure 3.** AIRS brightness temperature difference images for overpasses on 22 and 23 May 2011. (a) 03:35 UTC, 22 May. (b) 13:17 UTC, 22 May. (c) 04:17 UTC, 23 May. Negative values suggest absorption by $SO_2$ gas. The red coloured spot (positive temperature differences) seen on panels (a) and (b) situated near the  vent may be due to the ash-rich column.

depth) and passive microwave measurements can be used to detect large (mm-sized) volcanic particles. These passive measurements have been recently supplemented by an active space borne lidar that can provide much needed vertical information. The Cloud-Aerosol Lidar with Orthogonal Polarization (Caliop) on board the CALIPSO polar-orbiting platform provides vertical

structure information from backscattered light, along the sub-satellite point but has a long repeat cycle which inevitably means limited coverage for rapidly evolving, spatially limited events such as low to medium size volcanic eruptions. Table 1 provides a list of the satellite instrument data used in this study including the salient characteristics of these instruments.

**Table 1.** Satellite instruments used in this study to identify volcanic emissions. The instruments are: AVHRR–Advanced Very High Resolution Radiometer; MODIS–Moderate resolution Infrared Spectroradiometer; AIRS–Atmospheric Infra-Red Sounder; IASI–Infrared Atmospheric Spectroradiometer Interferometer; SEVIRI–Spin-stabilised Enhanced Visible and Infra-Red Instrument; OMI–Ozone Monitoring Instrument; Caliop–Cloud-Aerosol Lidar with Orthogonal Polarisation.

| Instrument/Platform | Spatial resolution ($km^2$) | Temporal resolution (hours) | Parameter | (A)ctive or (P)assive |
|---|---|---|---|---|
| AVHRR/Meto-A, B | ∼1.1x1.1 | ∼3 | Ash | P |
| MODIS/Terra/Aqua | 0.25x0.25–1x1 | ∼6 | Ash/$SO_2$ | P |
| AIRS/Aqua | ∼13x13 | ∼12 | $SO_2$/Ash | P |
| IASI/Metop-A | ∼10x10 | ∼12 | $SO_2$/Ash | P |
| SEVIRI/MSG-2 | ∼3x3–10x10 | 0.25 | Ash | P |
| OMI/Aura | ∼13x24 | 24 | $SO_2$/AAI | P |
| Caliop/CALIPSO | ∼0.1x0.3 | 16 days | Aerosols | A |

## 4.1 Cloud-top height

An estimate of the column height of eruptive material is critical to understanding the movement of ash away from its source. Initial estimates of the height of the ash column, based on radar measurements by the IMO (Petersen et al., 2012), suggested that it had reached ∼20 km. However, the satellite data analysed here, together with radiosonde temperature profiles made at Keflavik airport, indicate that the column maximum height did not exceed ∼16 km and varied between ∼8 km and ∼16 km from onset at 19:00 UTC on 21 May 2011 until a lowering sometime after 05:15 UTC on 22 May. The radar has a vertical resolution of between 2–5 km at a range of 75 km (Petersen et al., 2012, see. Fig.2), so a radar estimate of 15–25 km is broadly consistent with the satellite estimate. The measurements reported by Petersen et al. (2012) from radars situated at Keflavik (∼257 km away) and on a mobile platform show an oscillatory behaviour of the plume top on 21–22 May with a considerable drop in height to 10 km around 20:00 UTC on 22 May, followed by a drop to 6 km (or lower) at ∼11:00 UTC on 23 May, and another drop to less than 3 km at ∼16:00 UTC. The height does not exceed 8 km thereafter.

At the start of the  eruption a tall and optically thick column extended several kilometres into the atmosphere. During this phase, the large opacity of the cloud makes ash retrieval using infrared and ultra-violet radiation very difficult and it is likely

that the cause of the opacity is due to condensed water vapour and steam in the column. Particle sizes are large (mm to cm size) and hot gases, principally water vapour dominate the emissions. The evolution of the column can be studied using single-channel infrared measurements (Fig. 4), by making the assumption that the cloud is behaving as a blackbody and the brightness temperatures correspond to the cloud-top temperature. The blackbody assumption is generally quite good, but because the cloud

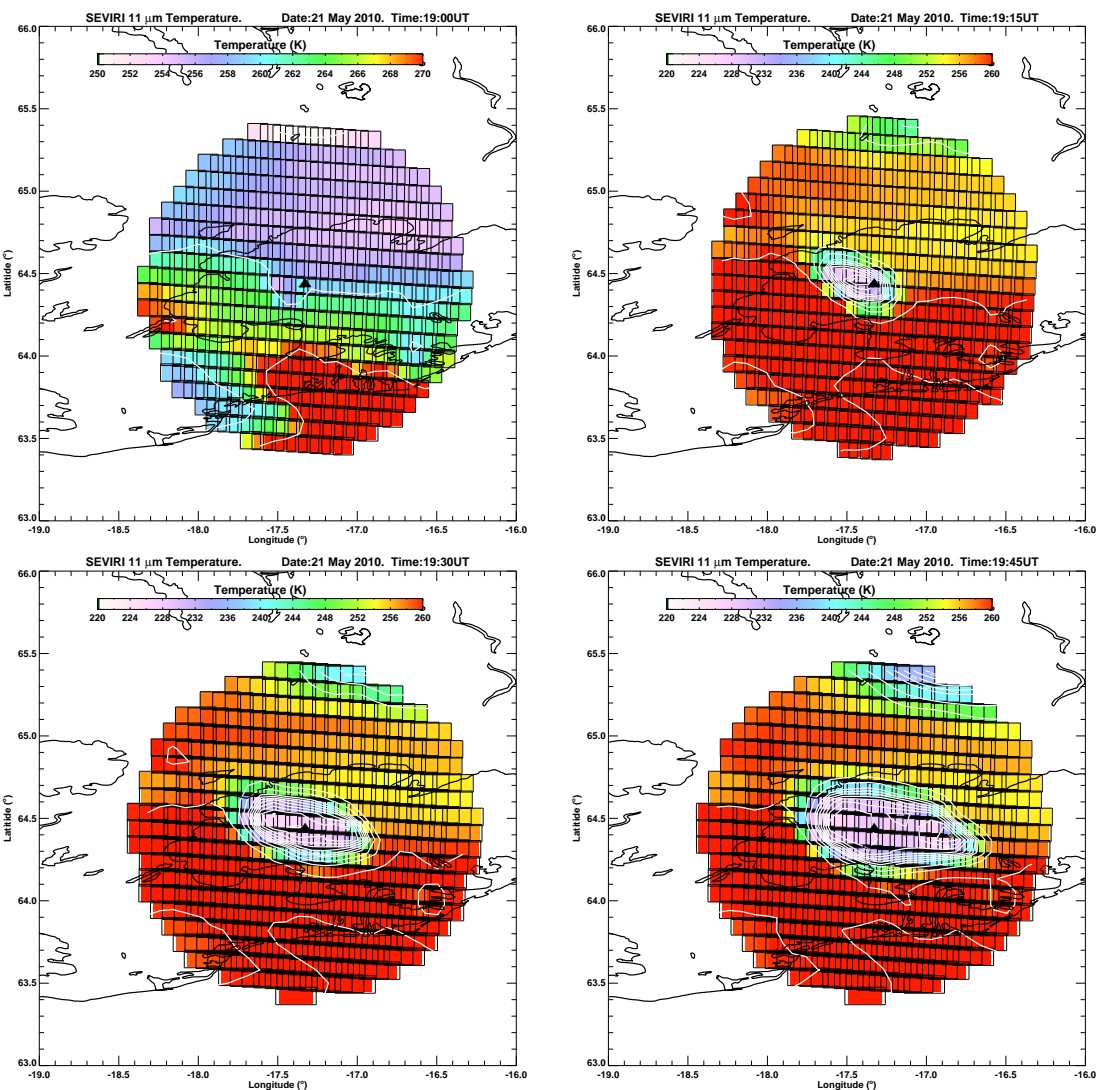

**Figure 4.** Close-up views of the 11 $\mu$m brightness temperatures from the SEVIRI instrument at 15 min intervals, starting at 19:00UTC on 21 May 2011. Isolines (contours) of brightness temperatures are shown in white to highlight the location and expansion of the top of the column. The rapid plume rise may be interpreted by the change in extent of the cloud top temperatures. There is no evidence of an eruption on the image at 19:00 UTC. The location of  is shown as a black triangle.

may overshoot, undercooling may occur (Woods et al., 1995) leading to cloud tops with IR temperatures many degrees Kelvin below the background atmospheric temperature. Radiosonde data from Keflavik airport (see map Fig. 2) were used to relate the IR brightness temperatures to cloud top height. The radiosonde data show a tropopause at ∼8.5 km and a dry layer between 3–4 km[2]. The winds are towards the south-west and south-south-west up to about 4 km, then westerlies up to the tropopause and high level winds from the south. A tall column of ash entering this highly sheared atmosphere will suffer transport in

at least three different directions. Plume evolution can also be estimated from spatial changes in the brightness temperature images and is a useful way to identify the onset of column growth.

The shadow cast by the eruption clouds seen on some MODIS images can also be used to estimate cloud top height (Prata and Grant, 2001). A MODIS/Aqua 250 m resolution image acquired at 13:15 UTC on 22 May shows the column with a strong shadow cast onto the ground and cloud below (Fig. 5). Utilising the geometry of the satellite and sun viewing directions and

the contrast difference between the dark shadow and brighter cloud/ice below, the highest parts of the ash column in this image are estimated to be 16±1 km, with other parts of the column having lower tops. It can also be seen that there is a tephra layer to the south of the column that appears to be detached and dispersing southwards. This layer may have arisen from a less vigorous phase of the eruption (when the column top was lower) or possibly was formed from ash rising off a PDC or from a partial column collapse. Whatever the exact mechanism involved, this low-level tephra plume is effectively independent of the

tephra fed into the upper column at source and could therefore be treated as a separate source for the purpose of forecasting its movement. The layer is evident on later MODIS images (see later Fig. 9) and it is this low-level tephra layer that eventually travels eastwards towards Scotland and on to south-western Scandanavia.

## 4.2   Spaceborne lidar measurements

The CALIOP instrument on board the polar orbiting CALIPSO platform is a polarization-sensitive, elastic backscatter lidar

capable of providing high vertical resolution (∼60 m) attenuated backscatter profiles of clouds and aerosols, and cloud-top heights. The instrument transmits linearly polarized light and measures the return signal at 532 and 1064 nm. The components perpendicular and parallel to laser polarization are measured separately at 532 nm. Details of the instrument, the science applications and an example of its use in a volcanic study may be found in Hunt et al. (2009), Vaughan et al. (2009) and Winker et al. (2012). The lidar is near-nadir pointing, has a ground footprint diameter of 70 m and a repeat time of 16 days, which

limits the number of times the lidar beam coincides with a target of interest. Ten CALIOP coincidences were identified for the ash and $SO_4^{2-}$ clouds between 21–23 May, 2011. Figure 6 shows an example of a CALIOP pass on 23 May when the CALIPSO trace intersected an ash cloud to the south of Iceland and a $SO_4^{2-}$ layer to the north. The upper left-panel shows indices based on coincident AIRS brightness temperature difference measurements, using an index to indicate ash (orange/red colours) or $SO_2$ (shades of blue). Details of the ash and $SO_2$ indices can be found in Prata et al. (2015) and Hoffmann et al.

(2014), respectively. The upper-right panel (Fig. 6b) shows a MODIS/Aqua true-colour image acquired at the same time as the AIRS measurements.Panel (c) shows the CALIOP attenuated backscatter signal measured at 532 nm. The black horizontal line indicates the height of the tropopause determined from GMAO (Global Modelling and Assimilation Office) reanalysis data

---

[2]These heights may be a little higher, ∼1–1.5 km over the high terrain of the Vatnajökull glacier.

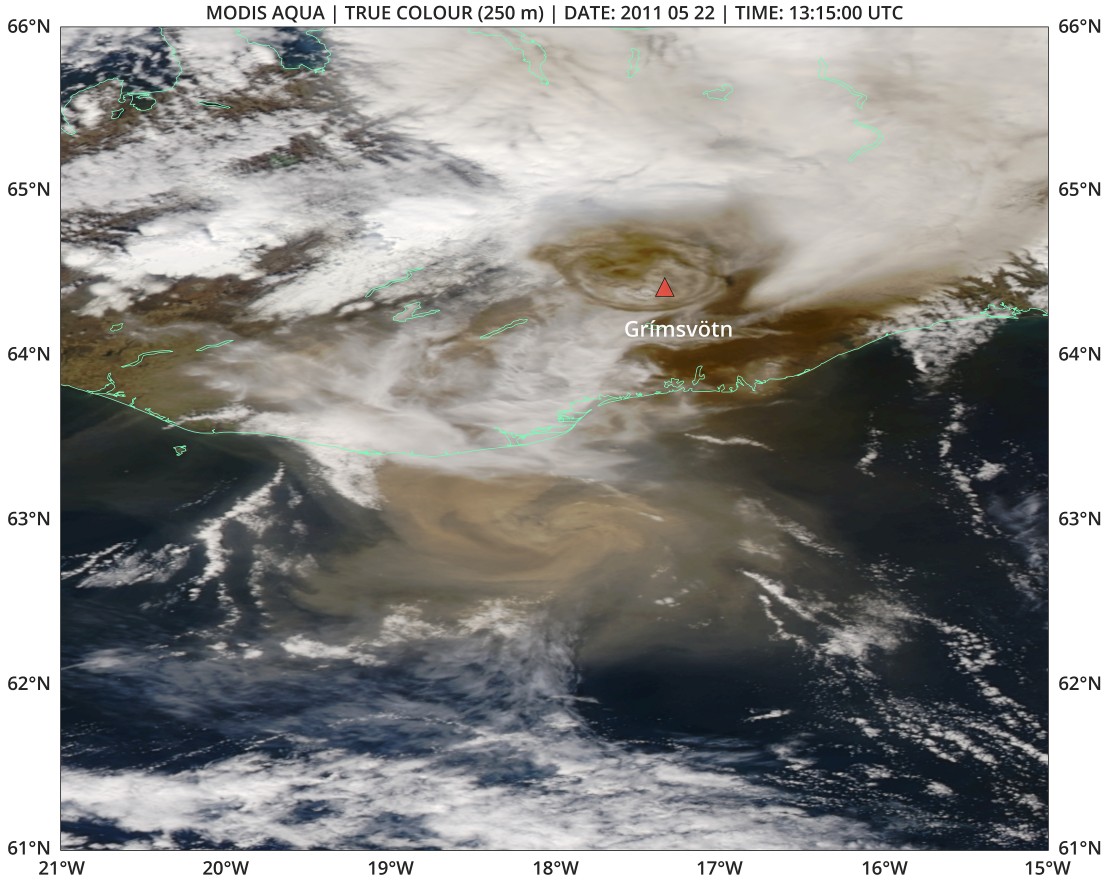

**Figure 5.** MODIS true-color 250 m resolution image of the eruption column, showing the shadow cast on the ground and cloud below (to the N of the column). Note also the the ash layer off the south coast that appears detached from the main column, suggesting that it is no longer being fed by ash from the vent. *Image: MODIS/Aqua, 22 May 2011, 13:15 UTC.*

(Rienecker et al., 2008). The strips at the base show collocated AIRS pixels along the CALIOP track where ash and $SO_2$ have been identified. Between $\sim$59.9°N and $\sim$62.7°N (left-most white-coloured ellipse) a tropospheric ash cloud is detected in the AIRS data and the CALIOP backscatter signal suggests these cloud layers have heights of $\sim$1–6 km. Between $\sim$68.6°N and $\sim$72.0°N (right-most white-coloured ellipse) a stratospheric cloud layer of $SO_2$ is detected in the AIRS data. The CALIOP instrument is insensitive to $SO_2$ but does scatter light from ash and $SO_4{}^{2-}$ aerosols as well as meteorological clouds of ice and water droplets. The height of this layer in the CALIOP curtain is between 10–12 km and above the tropopause. The lower-right panels show vertical profiles of the backscatter for these two layers, averaged over the two latitude sections identified. These data suggest that the upper layer is most likely to be an $SO_4{}^{2-}$ layer coincident with the $SO_2$ gas. Low volume-depolarization ratios ($\delta_v\sim$0.1–0.2), indicative of spherical particles, within the stratospheric layer are also consistent with a $SO_4{}^{2-}$ layer rather than ash or ice clouds. For the eruption of Sarychev Peak, Prata et al. (2017) found a mean $\delta_v$ of 0.05$\pm$0.04 and for Kasatochi

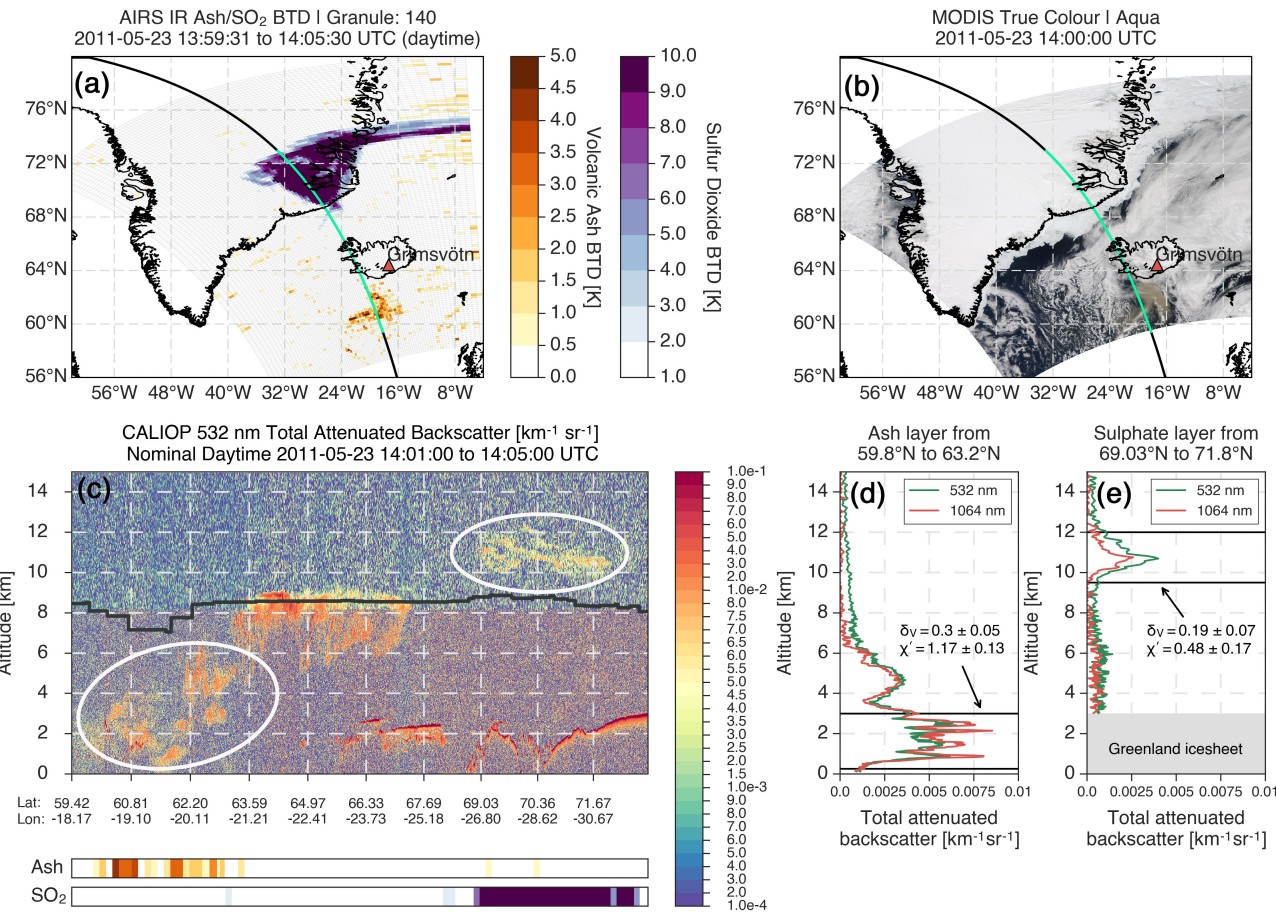

**Figure 6.** *Top-panels*, (a): AIRS brightness temperature difference for volcanic ash (yellow/orange/red) and AIRS brightness temperature difference for SO₂ (shades of blue). The ascending trace (travelling from South to North) of the CALIPSO satellite is indicated by the black/green line, where the green-colored portion of the line indicates the region of overlap between the Caliop and AIRS image data. (b) MODIS/Aqua true-colour image showing the low-level ash cloud (brown). *Bottom-panels:* (c) Caliop backscatter curtain for 532 nm light with the ash and sulphate layers indicated by the white-coloured ellipses. The lower strips show temperature differences based on the AIRS data indicating regions affected by ash and SO₂ gas. Panels (d) and (e) show vertical profiles of backscatter for ash and $SO_4{}^{2-}$, respectively. $\delta_v$ is the volume depolarisation ratio and $\chi'$ is the colour ratio. The horizontal black lines in panels (d) and (e) show the height range over which the parameters have been calculated. Date and time of overpass: 23 May 2011, 14:01–14:06 UTC.

10    $\delta_v$ was 0.08±0.03. However, these were based on nighttime measurements and daytime data are noisier in the backscatter signal and can contribute to higher than expected $\delta_v$ values. For Puyehue-Córdon Caulle (dominated by ash particles), $\delta_v \sim$ 0.28±0.03. The threshold between sulphates and ash used was $\delta_v \sim$0.2. This makes the observation somewhat ambiguous.

There are three potential interpretations:

(1) The layer was sulphate and the $\delta_v \sim 0.2$ was due to daytime noise in the backscatter signal.

(2) The layer was sulphate with some depolarisation from ice particles (not very likely in the stratosphere).

(3) The layer was sulphate with some depolarisation from small ash particles (below the detection limit of AIRS).

The last interpretation is certainly consistent with observations of some small amounts of ash in the northern part of the plume (Prata and Rose, 2015).

The colour ratios ($\chi'$;see https://eosweb.larc.nasa.gov/PRODOCS/calipso/Quality_Summaries/CALIOP_L2LayerProducts_3.01.html for more details of these parameters) are $\chi' \sim 1$ for ash and $\chi' \sim 0.5$ for $SO_4^{2-}$, showing a clear difference between these two aerosol layers. The ash layer shows considerable vertical structure, with thin layering evident below 3 km and a broader feature peaking $\sim 4.5$ km. These data provide compelling evidence for ash and $SO_2$ (and $SO_4^{2-}$) separation from the eruption, and also provide quantitative estimates of the height separation with a lower troposphere ash cloud and a stratospheric gas and aerosol cloud.

### 4.3  SO$_2$ gas

The satellite instruments used to retrieve $SO_2$ are also shown in Table 1. Details of the retrieval algorithms may be found in the papers by Prata and Bernardo (2007) for the Atmospheric Infrared Sounder (AIRS), by Yang et al. (2007a) for the Ozone Monitoring Instrument (OMI) and Clarisse et al. (2008) for IASI. AIRS data provides an excellent view of the $SO_2$ dispersion; see the Supplemnatry figure S2. $SO_2$ was first detected in AIRS data at 03:35 UTC on 22 May, which was the first overpass of the Aqua satellite platform over Iceland following the initial  eruption. A large cloud of $SO_2$ gas was detected over the Vatnajökull glacier, slighly displaced to the north of . In subsequent AIRS overpasses the $SO_2$ cloud grew larger and spread predominantly northwards, reaching the Greenland coast by 04:17 UTC on 23 May, $\sim 12$ hours later. The $SO_2$ cloud then spread westwards and eastwards while still propagating northwards into a long filament. The $SO_2$ layer height cannot be inferred directly from the AIRS retrievals, but the direction of travel and transport modelling suggests a height of $\sim 8$–10 km; which implies the $SO_2$ was stratospheric. The mass of upper troposphere lower stratosphere (UTLS) $SO_2$ calculated from the AIRS data is shown in Figure 7. The maximum $SO_2$ mass was found to be $\sim 0.24 \pm 0.05$ Tg at 14:00 UTC on 23 May, 2011. Although the AIRS retrievals are only strictly valid for the UTLS, in this case it is most likely that the majority ($>90\%$) of the $SO_2$ was located in the UTLS. Identification of a volcanic layer in Caliop lidar data was difficult initially, suggesting that conversion to $SO_4^{2-}$ aerosol was not yet sufficient to provide a good signal and that few ash particles were collocated with the $SO_2$.[3] At least three other satellite sensors detected the high level $SO_2$ cloud: OMI on the Aura platform, GOME-2 on MetOp-A and IASI also on MetOp-A. Table 2 shows estimates of the daily $SO_2$ mass from each of the sensors. OMI observations are

---

[3]The Caliop lidar is insensitive to $SO_2$ gas, but backscatter depolarisation and colour ratio values from both $SO_4^{2-}$ and ash particles can often be identified for strong layers.

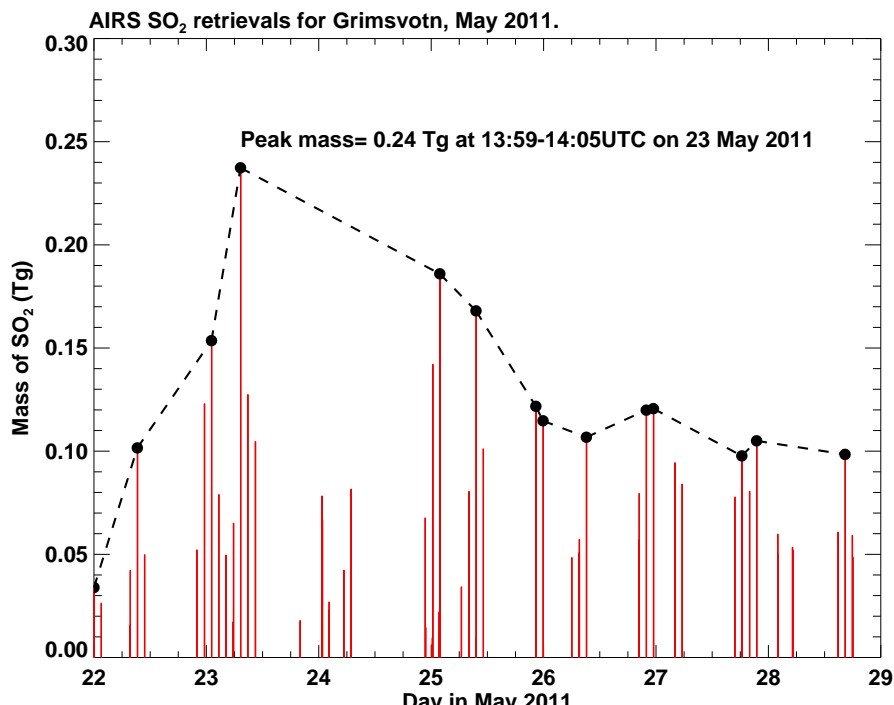

**Figure 7.** AIRS UTLS $SO_2$ mass loading (Tg) as a function of time for 22–29 May, 2011. The dashed line shows the locus of maximum mass loadings – since AIRS sometimes has incomplete coverage of the whole plume a true estimate of the maximum $SO_2$ mass is difficult.

shown in the Supplementary Figure S3. Sigmarsson et al. (2013) estimated the sulfur budget for the eruption and made use of satellite $SO_2$ measurements.

Although there is some disparity between the estimates from the different sensors, when the effects of differences in height sensitivity, timing differences, field-of-view sizes and swath overlap are taken into account, the values fall within the expected error bounds. The means and standard deviations for 22, 23, 24, 25 and 26 May are, 0.128±0.08 Tg, 0.238±0.09 Tg, 0.200±0.08 Tg, 0.183±0.07 Tg, and 0.180±0.06 Tg, respectively. We therefore conclude that between 0.13–0.24±0.1 Tg $SO_2$ was released into the UTLS by during the period 22–26 May, 2011, about half the total estimated amount released to the atmosphere. Carn *et al.* (2016) estimated a maximum $SO_2$ loading of ∼0.38 Tg and Sigmarsson et al. (2013) estimated ∼0.2 Tg(S) or ∼0.4 Tg($SO_2$).

### 4.4 Ash

Volcanic ash retrievals were performed using the methods outlined by Wen and Rose (1994) and Prata and Prata (2012). Data from MODIS, AIRS and IASI, all on polar-orbiting platforms were used to determine brightness temperatures and ultimately fine (effective radii < 16 $\mu$m) ash mass loadings and particle sizes. Geostationary data from SEVIRI provided measurements

**Table 2.** SO$_2$ total mass estimates from four different satellite instruments (two infrared and two ultraviolet) from 22–26 May 2011. [1]L. Clarisse, private comm. [2]A. Richter, http://www.iup.uni-bremen.de/scia-arc/

| Instrument | Date in May, 2011 | | | | |
|---|---|---|---|---|---|
| | 22 | 23 | 24 | 25 | 26 |
| | Total mass (Tg) | | | | |
| AIRS | 0.10 | 0.24 | 0.18 | 0.12 | 0.11 |
| IASI[1] | 0.23 | 0.32 | | | |
| OMI | 0.15 | 0.28 | 0.29 | 0.25 | 0.20 |
| GOME-2[2] | 0.03 | 0.11 | 0.13 | 0.18 | 0.23 |

every 15 minutes from which brightness temperatures in 5 infrared channels could be used to detect and quantify the very fine ash component. Figure S4 (Supplementary) shows ash mass and effective particle size retrievals from SEVIRI at 6 hourly intervals on 23 May 2011.

The mass of very fine ash was estimated using SEVIRI images by averaging in hourly intervals (4 estimates per hour) and adjusting the estimates for changes in viewing angle, which can cause an error in the cloud top temperature estimation. Mass is estimated by identifying only ash affected pixels (using a sequence of cloud tests), multiplying these by the area of the pixel (which varies with scan position) and summing them to arrive at a total mass. For this case, the zenith viewing angle decreased from ∼60° to ∼10° as the ash cloud progressed eastwards.

The maximum mass of very fine ash was estimated to be 0.19±0.03 Tg late on 23 May. This is about 0.05% of the total mass of magma erupted, suggesting that the very fine ash fraction is <<1 wt% of the overall mass of tephra produced by the eruption. Four MODIS overpasses were also used to estimate very fine ash mass, shown in Fig. 8 together with estimates from IASI (Clarisse, private communication; also see Moxnes et al. (2014)) that are sampled twice per day. The MODIS retrievals are shown in Figure S4 (Supplementary material).

The MODIS data give slightly higher estimates than SEVIRI, decreasing from ∼0.23 Tg at 12:05 UTC on 23 May to ∼0.15 Tg at 03:25 UTC on 24 May. The low SEVIRI estimates at the start of the series are a consequence of the inability of the SEVIRI retrieval scheme to quantify ash at these high zenith viewing angles and the confounding effects of meteorological cloud that sometimes overlaid the ash (the ash layer was mostly confined to heights below ∼ 3 km; see also Fig. 9). IASI retrievals are consistently higher than SEVIRI and MODIS until late on 24 May when there is closer agreement. Prata and Prata (2012) showed that the SEVIRI mass loading retrievals were consistent with ground level PM10 concentration measurements at several stations in Scandanavia (e.g. at Bergen and Oslo), if the ash cloud was assumed to be confined to a layer no deeper than 3 km. The reason for the large differences between IASI and SEVIRI retrievals is under investigation, but IASI has a greater sensitivity to ash due to the higher spectral resolution, and the assumptions used in the retrievals are different (Clarisse and Prata, 2016). It is also clear from some of the MODIS images that meteorological cloud overlaid the ash cloud and it is difficult to retrieve ash from these broadband low spectral resolution data under these circumstances.

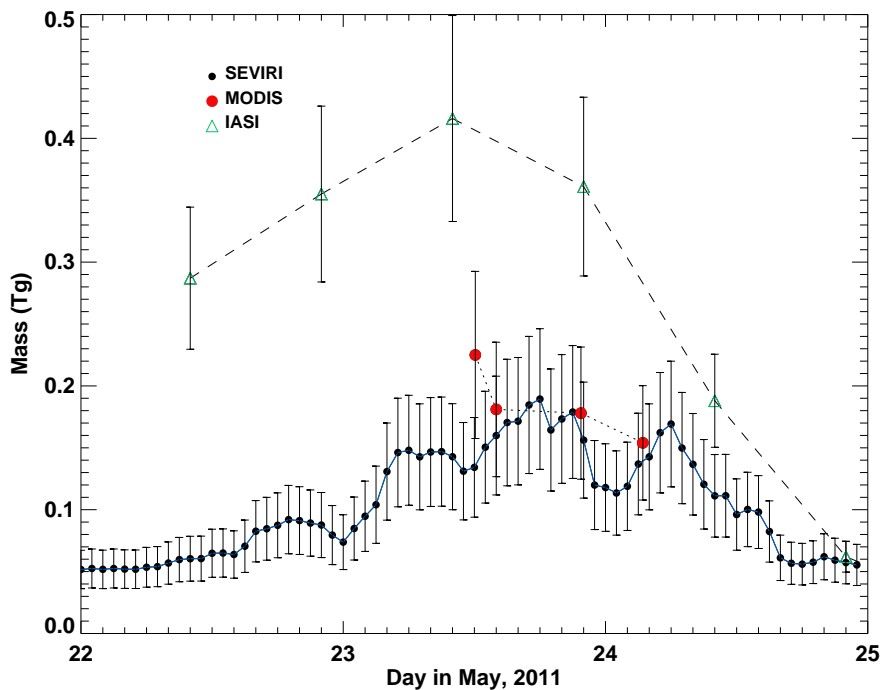

**Figure 8.** Very fine ash mass (Tg) estimated from SEVIRI, MODIS and IASI data for the period 22–24 May, 2011.

## 4.5   Error in ash retrievals

The error (precision) in estimating very fine ash mass from infrared retrievals has been investigated by Wen and Rose (1994) and Prata and Prata (2012), who suggest errors of 40–50%. Stevenson et al. (2015) discuss potential errors in satellite retrievals by using cryptotephra data to speculate that larger particles exist in dispersing ash clouds (although no atmospheric observations are presented) and claim through modelling studies that current retrieval schemes (all of them) underestimate mass loadings

5   because of the dense sphere assumption and lack of sensitivity to particles with diameters $> 10$ $\mu$m. Estimating precision in retrievals is difficult because of the uncertainties in the input parameters, such as the complex index of refraction, the size distribution and the shapes of the particles, although shape is generally found to result in the smallest discrepancy of the input parameters with theoretical simulations showing differences in the range of 10–40% (Yang et al., 2007b; Kylling et al., 2014). An additional problem with estimating precision due to shape is that apart from having no observations, the effect of

10   their statistical orientation in space and the distribution of the shapes as a function of particle size is unknown and potentially large. Attempting to model these uncertainties in the absence of any observational constraints is unproductive. An alternate strategy, and one that is adopted widely, is to use what little data there is and compare retrievals with independent observations to estimate the accuracy with respect to the independent estimate. It is acknowledged that this approach may be misleading because uncertainties in assumed parameters may cause errors to cancel and lead to better results than otherwise expected.

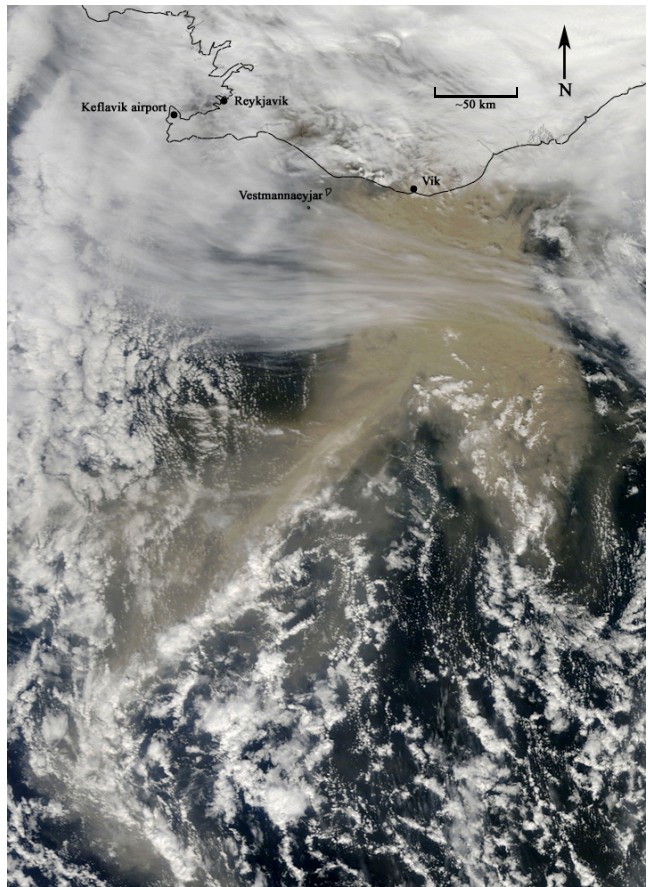

**Figure 9.** MODIS true-color image showing the low-level ash cloud coming off Iceland and spreading southwards. Notice that meteorological cloud is clearly evident above the ash layer and is either obscuring the ash layer over Iceland, or the ash layer ends near the coast.  is just off the image to the north. (*Image: MODIS/Terra, 23 May 2011, 12:05 UTC.*)

Nevertheless, with the limited independent observations available, accuracies in mass loadings appear to be in the range of 20–50%. Clarisse and Prata (2016) discuss errors (precision and accuracy) in ash retrievals and suggest areas where more research is needed. In this study, we focus only on the accuracy in the retrievals for the  eruption. Uncertainties that are identified due to cloudiness, lack of thermal contrast (either ash that is either optically too thick or optically too thin), radiometric errors and estimates of cloud-top and surface temperature are included in the error budget. In the case of the ash retrievals for , the error estimates are within the expected range, giving an error of $\pm$0.1 Tg or roughly 20–50% of the estimated mass of very fine ash. It is emphasised that this is not the total mass emitted by the volcano which is typically a few percent of the total mass. It is however, the mass fraction that is dispersed by the winds and the very fine ash that can cause damage to aircraft jet engines. Individual mass loading errors can be lower than 20% and also much higher, depending mostly on contamination of the pixel

by meteorological cloud, but generally these are not validated because there are no independent measurements of mass loading. IASI retrievals have a precision also in the range of 20–50% but their accuracy is unknown as no independent validation has been done. IASI retrievals were biased high compared to the SEVIRI and MODIS retrievals in this case and the cause is not yet understood.

Retrieval methods are being continually improved and there is an international effort (http://cimss.ssec.wisc.edu/meetings/vol_ash15/) to inter-compare retrieval schemes and help reduce uncertainty. At the current time no firm conclusions have been made about retrieval accuracy as no robust validation has been made. Uncertainties can only be assessed against independent observations and so far independent measurements of mass loading are extremely sparse, let alone independent measurements of atmospheric ash particle size distributions, shapes and composition.

Tesche et al. (2012) and Ansmann et al. (2012) report lidar measurements of ash mass concentrations in the range 100–340 $\mu$g m$^{-3}$. Moxnes et al. (2014) report values $< 100$ $\mu$g m$^{-3}$ based on aircraft data and modeling. These data, our data, previous measurements from Eyjafjalljökull (lidar, airborne and ground-based air quality) all provide adequate support for the assumptions used in satellite-based IR retrievals. The error estimates for the   eruption used here are robust, but should not be extended to all ash retrievals or for any other eruption.

We estimate that the amount of ash transported towards Europe between 22–25 May, 2011 was 0.2–0.4±0.1 Tg (very fine ash). By comparison, Stohl et al. (2011) estimated 8.3±4.2 Tg of very fine ash emitted during the Eyjafjallajökull eruption in April–May, 2010, which is an order of magnitude greater from an eruption that was a factor ∼2 smaller in total erupted mass than the  eruption. Moxnes et al. (2014) estimate that a total of 0.49±0.1 Tg of very fine ash from , based on modelling results, that utilised IASI retrievals not used in our study.

## 5   Possible column collapse and PDCs

The lower level ash plume was beginning to form at 19:15 to 19:20 UTC and was fully developed by 20:00 UTC on 21 May. Ground-based observations (see Supplementary photographs) show that vent from  to Blágil in the Laki area is about 60 km and the lower level ash plume reached there in about 1 hour. The MODIS satellite data shows that the low-level ash layer ($<$6 km high), was present off the south coast of Iceland on the morning of 22 May and is also clearly observed 24 hours later (see Fig. 9). This layer appears to be detached from the main eruption column. Photographic evidence (see Fig. 1, lower panel) shows a shallow ash cloud/plume[4] at low level surrounding the main column (a 'skirt'), and another plume-like ash-rich layer higher up and at about half the height of the column. These observations suggest the possibility that the column may have undergone partial collapse sometime during the evening of 21 May, causing an outflow of ash, not dissimilar to the outflow often observed from a collapsing thunderstorm. As large ash aggregates fall through the column, enhanced by the presence of copious amounts of water, for example see Telling et al. (2013b) for a discussion of this process, ice would have formed on the ash, increasing the size and fall speed and effectively removing particles from the column. These ice-coated ash aggregates,

---

[4]We define an ash cloud as an identifiable structure wholly disconnected from the vent, whereas an ash plume has an identifiable connection to the source vent.

sometimes termed volcanic hail would have fallen out of the cloud very rapidly. The process of ash falling through the column would have caused compression of the lower part of the column, and a mechanism for driving a gravity current of ash outwards from the column. Such PDCs may have supported plumes with ash rising from the regions immediately outside the vent area. The light, SW winds in the lower troposphere favour propagation of the outflow towards the west, as observed, but it is likely that the ash formed a 'skirt' surrounding the collapsing column. Column collapses can also cause pyroclastic density currents,

5 so the two mechanisms may not be seen as separate. A schematic of the proposed processes is shown in Fig. 10.

The speculation that a partial collapse and/or generation of an ash skirt (PDC without collapse) moving outwards from the plume is supported by the photographs shown in Fig. 1 and the MODIS satellite image shown in Fig. 5. Jude-Eton et al. (2012) showed that PDCs occurred during the 2004 eruption (see the photographs in their Fig.2(a) and (b)). In these instances a column collapse is not required; their generation results from the development of a collapsing veil of material Carey and Bursik

10 (2015). The PDCs emerge from the veil and follow the underlying topography. Either way the phreatomagmatic nature of the eruption with the injection of large amounts of water appear to be important ingredients leading to the observation of a skirt of ash propagating at lower levels. The outflow from this mechanism may have been relatively fast; the AIRS satellite observations

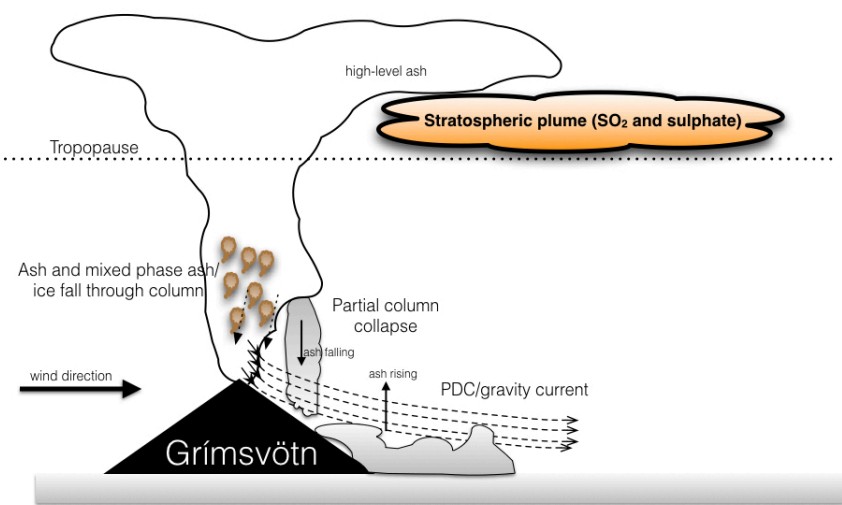

**Figure 10.** A schematic of the principle features of the eruption column. Large hydrometeors composed of ash aggregates and mixed phase ash/ice particles fall through the column competing with the upward force of the eruption, eventually causing the column to collapse, the generation of one or more PDCs pushing an outflow of ash-rich air into the lower troposphere. Ash rises from the PDC and falls in the collapsing column. A high level plume of $SO_2$ with some ash penetrates the tropopause. The height range of the gravity current (or pyroclastic density current) is unknown but likely to extend from the ground to the top of the observed ash skirt.

suggest the column had stopped rising by 04:17 UTC on 23 May. Our photographic series show that the plume stopped rising

in the time frame 19:30–20:00 UTC on 21 May. It stayed relatively elevated, i.e. between 15-19 km and according to daily observational reports it stayed at this level until mid-morning of 22 May. On the same day by noon it had dropped below 10 km and stayed there through the 23 May. At the end of that day it dropped below 5 km, and more or less remained below that height for the rest of the eruption. The low-level ash layer persisted close to the south coast of Iceland for at least 24 hours before starting its journey further southwards and then eastwards. Atmospheric transport processes (e.g. buoyant transport, advection
by the low level winds, particle settling) are acting on this ash cloud, but the low-level winds were not strong and so the ash moved slowly. The cloud may also have been fed by new ash from the on-going minor eruptions.

The ash transported southwards from which begins within the first hour of the eruption arises not directly from the emissions at the vent but most likely from a possible partial collapse of the eruption column which can no longer be sustained, or from the generation of one or more PDCs. The southward movement of the ash 'skirt' can best be seen in the MODIS image acquired
on 23 May 2011 at 12:05 UTC (Figure 9). The ash mass retrievals for this image (and three later images) are shown in Figure S4 (Supplementary Material) (top-left panel) where three mass loading levels are indicated: 0.2, 2 and 4 g m$^{-2}$.

Further support for rapid removal of ash before transport is provided in the photograph shown in Figure 11 taken on 31 May, just 8 days later on the Vatnajökull glacier near . The photograph shows a short vertical section dug into the deposit with evidence of hail. The presence of hail within the deposit has also been described by Arason et al. (2011), who found
hailstones of 1–2 mm size infused with ash. Gudmundsson (2013) has estimated the amount of water melted by the eruption and since no Jökulhups were observed it may be assumed that much of that water went into the plume, in the form of hot water vapour (steam) and also contributing to ice and hail formation within the column. A large amount of lightning was observed in the eruption column and clouds, also suggesting the presence of hydrometeors. It is difficult to estimate whether the column

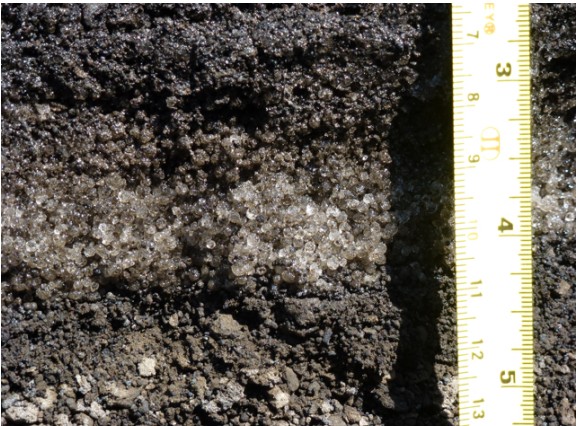

**Figure 11.** Photograph of a vertical section taken on the Vatnajökull glacier at a location where there was significant ash fall from , on 31 May 2011. There is evidence of millimetre-sized hail in the deposit. Photo taken by Adam Durant during visit organised by Fred Prata.

collapsed more than once but there does seem to be evidence that an ash surge existed on the morning of 22 May. A MODIS image acquired at 05:15 UTC on 22 May (∼10 hours after the start of the eruption) appears to show gravity waves emanating from the column, and a skirt of ash spreading southwards and then curving around the north-eastern coast of Iceland. The photographic evidence suggests that the process started much earlier. These waves could have been formed when the column sloughed, causing a cold ash surge driven by the buoyancy force due to the vertical gradient in the density. Figure 12(a) shows
5   the 250 m resolution MODIS band 2 (841–876 nm) image reprojected, calibrated to reflectance and digitally enhanced to highlight various features.

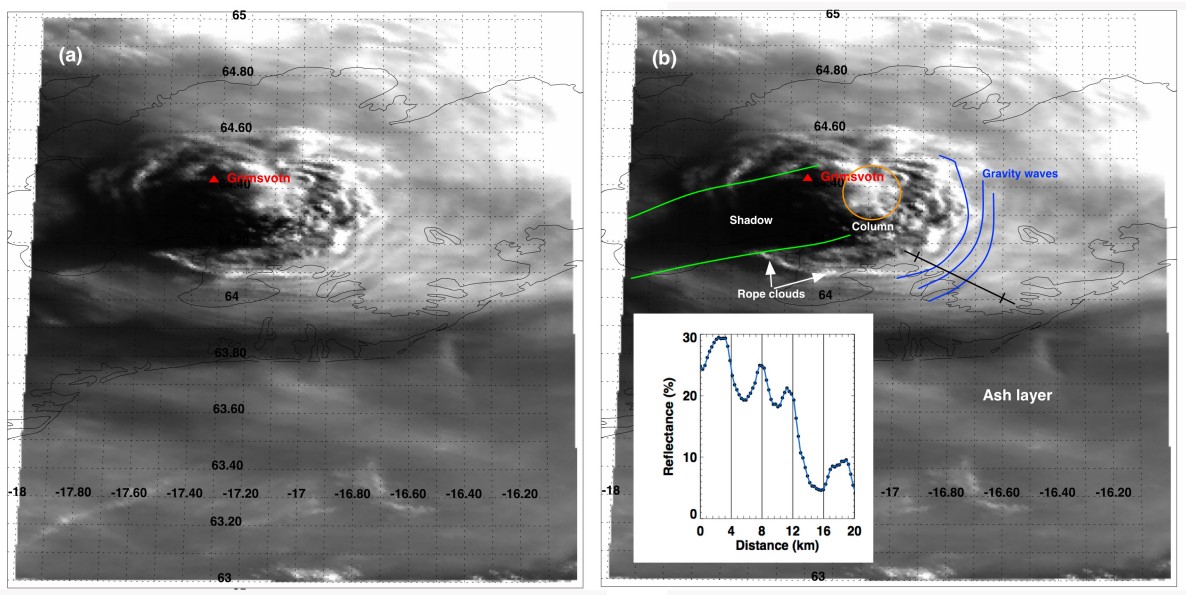

**Figure 12.** (a) MODIS/Aqua 250 m resolution 841–876 nm reflectance image showing the ash column, ash column shadow, ash layer, rope clouds and gravity wave features. (b) Annotated version of (a) with inset plot showing reflectance along the a line (indicated in black) from the ash column towards the south-east. The blue-coloured arcs indicate the locations of wave-like fronts, the orange-coloured circle indicates the outline of the ash column, which is casting a strong shadow (westwards of the column) on the underlying clouds of ash and water clouds. The apparent wavelength of the waves is ∼4–6 km. The solar zenith and azimuth and MODIS viewing zenith and azimuth angles close to the column are: 82.5°, 56.4°, 55.5° and −53.4°, respectively. Image acquired at 05:15 UTC on 22 May 2011.

    The features are identified as the eruption column (slightly east of the volcano location, indicated by the red triangle), its shadow cast westwards onto a lower layer of ash and meteorological cloud, an ash layer extending along the south coast of Iceland, rope clouds, and gravity waves (Fig. 12b). The reflectance (in %) along a transect indicated by the black line is
10   also shown inset in Fig. 12(b). Variations in reflectance along the transect occur due to height variations in the cloud and the solar/sensor viewing geometry. The approximate wavelength of the waves is ∼4–6 km.

# 6 Insights on the mechanisms and conditions for ash separation from a plume

The observations provide strong evidence that separation of ash occurred predominantly in the convectively-rising part of the eruption column, where the motion is driven by a buoyancy force arising from a density difference between the column and the atmosphere, rather than at the source or in the laterally intruding ash cloud. Figure 1 shows convincing evidence that separation is occurring at the convective column. The buoyant volcanic plume is a complex physical environment, with multiple interacting

phases, highly turbulent flow fields, and coupled nonlinear physical and chemical processes occurring. Despite this complexity, much insight into the dynamics of volcanic plumes has been gained from mathematical models of turbulent buoyant plumes (Morton et al., 1956) which have been extended to model thermodynamics and transport of solids in volcanic plumes (see e.g. Wilson et al., 1978; Sparks, 1986; Woods, 1988; Glaze and Baloga, 1996; Sparks et al., 1997a; Bursik, 2001; Woodhouse et al., 2013).

Here we use an integral model of volcanic plumes to gain insight into the physical processes that could lead to an abrupt separation of ash from the plume at . We adopt the integral model of Woodhouse et al. (2013) which includes descriptions of the thermodynamics of phase changes of water, the effect of atmospheric winds on the plume dynamics, and detailed profiles of the atmospheric structure during the eruption. Additional details of our modelling approach are given in the Appendix and a derivation of the system of equations adopted in our model are given in §2 and §3 of Woodhouse et al. (2013).

Our hypothesis is that the separation of ash from the convectively-rising plume that was observed at high altitude was due to rapid aggregation of ash particles, mediated by a rapid condensation of water in the plume. The presence of (liquid) water is likely to promote the aggregation of ash particles by allowing the formation of liquid bridges between grains (Brown et al., 2012; Van Eaton et al., 2012). The capillary forces in the liquid connections are much stronger than electrostatic attractions between dry grains (James et al., 2003), and therefore it is possible that wet aggregates can endure a collision which would

cause dry aggregates to break apart. Aggregation in the presence of liquid water or ice is extremely efficient, with aggregation time scales less than one-tenth of a second (Veitch and Woods, 2001; Costa et al., 2010). Costa et al. (2010) demonstrate that a particle size distribution that initially has a peak number density at $10\,\mu$m can evolve to a produce a peak in the number density at $100\,\mu$m in 60 seconds in an environment with condensed water available.

Because there is not that much very fine ash in the column to begin with to generate a sector wide plume collapse we cannot

be sure that aggregation is the sole driver. The particles in the $100\,\mu$m size fraction contain less than 10% of the mass erupted at any one time, so that even if all of this ash forms aggregates, the mass fraction is still small compared to the total mass. In the proximity of the volcano the tephra contains abundance of lapilli size clasts (2–64 mm in diameter), and over 50% of the proximal tephra is lapilli, and the fallout units are over 80% lapilli (i.e. 2–64 mm clasts ). Only the surges that generate the PDC deposits contain substantial amount of ash; more then 90%, because they do not have the capacity to carry the lapilli

clasts to start with. However, most of the tephra deposited by the PDCs is in the 0.1 to 1 mm (100–1000 $\mu$m) range, or >70%. Observations on accretionary lapilli (i.e. ash aggregates), indicate that this size range is too big to partake in ash aggregation by capillary forces plus electrostatic forces. The separation of the very fine ash, moving laterally and deposited from a laterally

moving current, and the lapilli size material that is processed vertically by being transported upwards and then falling out is not fully understood.

Rather than modelling aggregation explicitly, which is subject to great uncertainty, here we use a model of volcanic plumes to investigate whether conditions in the plume are favourable for wet aggregation and an abrupt fallout of solids. The maximum elevation of solid particles of a given size can be estimated by balancing the average vertical velocity of gaseous phases in the plume with the settling-speed of a particle (see the appendix). This provides a simple, yet robust, method of examining the consequence of aggregation; the estimated maximum fallout height is determined only by the particle size and the evolution of the particle size distribution is not required. Figure 13 illustrates a typical prediction obtained from our model for the plume from at 05:00 UTC on 22 May 2011.

Gentle winds and the large mass flux of erupted material result in a sub-vertical plume that is little affected by the wind. The model identifies an abrupt transition in the plume from dry conditions at lower levels (below approximately $10\,\mathrm{km}$) to an environment with a substantial amount of condensed water, and the low atmospheric temperature results in a predominance of ice with peak concentration in excess of $4\ \mathrm{g\,kg^{-1}}$ (Fig. 13b). The critical fall-out velocity of $50\,\mu\mathrm{m}$ particles is reached at an altitude of $18\,\mathrm{km\,asl}$ (Fig. 13c), above the neutral buoyancy height (Fig. 13d) and so these small particles could be carried to the plume top and into the lateral intrusion. Similarly, $100\,\mu\mathrm{m}$ particles could be transported to $17\,\mathrm{km\,asl}$. However, if these fine particles aggregate during their transport to produce larger grains, the velocity in the upper plume could be insufficient to support them. The velocity of the plume in the region above the condensation level falls below $5\ \mathrm{m\ s^{-1}}$ and therefore fine particles will take several minutes to reach the plume top; ample time for aggregation to occur (Veitch and Woods, 2001; Costa et al., 2010). For particles of diameters $500\,\mu\mathrm{m}$ and $1\,\mathrm{mm}$ the critical fall-out velocity is reached at altitudes of $11\,\mathrm{km\,asl}$ and $6.8\,\mathrm{km\,asl}$ respectively (Fig. 13c), which is below the neutral buoyancy height.

The small diameter grains carried above the condensation height are transported further (circulating in turbulent eddies) in an environment conducive to rapid wet aggregation, and therefore we expect the particle diameter to increase substantially. If aggregates grow sufficiently rapidly they will fall out before reaching the neutral buoyancy height. Deposits of tephra on the Vatnajökull glacier show evidence of hailstones infused with ash with diameters as large as $1$–$2\,\mathrm{mm}$. Aggregates of this size would readily fall from the plume and would be unlikely to be re-entrained into the plume due to their size and the width of the plume at the height at which fall out occurs. Smaller aggregates will also fall out of the plume but may not reach the ground proximal to the eruption column, instead being transported laterally by the wind.

## 7   Conclusions

The vertical separation of gases and particles in volcanic eruption columns occurs frequently and if it occurs in the presence of wind shear it is inevitable that this results in a lateral separation of gases and particles distally. Wind shear is ubiquitous and significant when eruption columns extend to the tropopause and consequently it should be expected that some separation will occur. Since also gases and particles are not always released in unison, the time-varying nature of the wind fields might also lead to separation even for a steady, low level eruption column. Gases and particles also separate within the column due

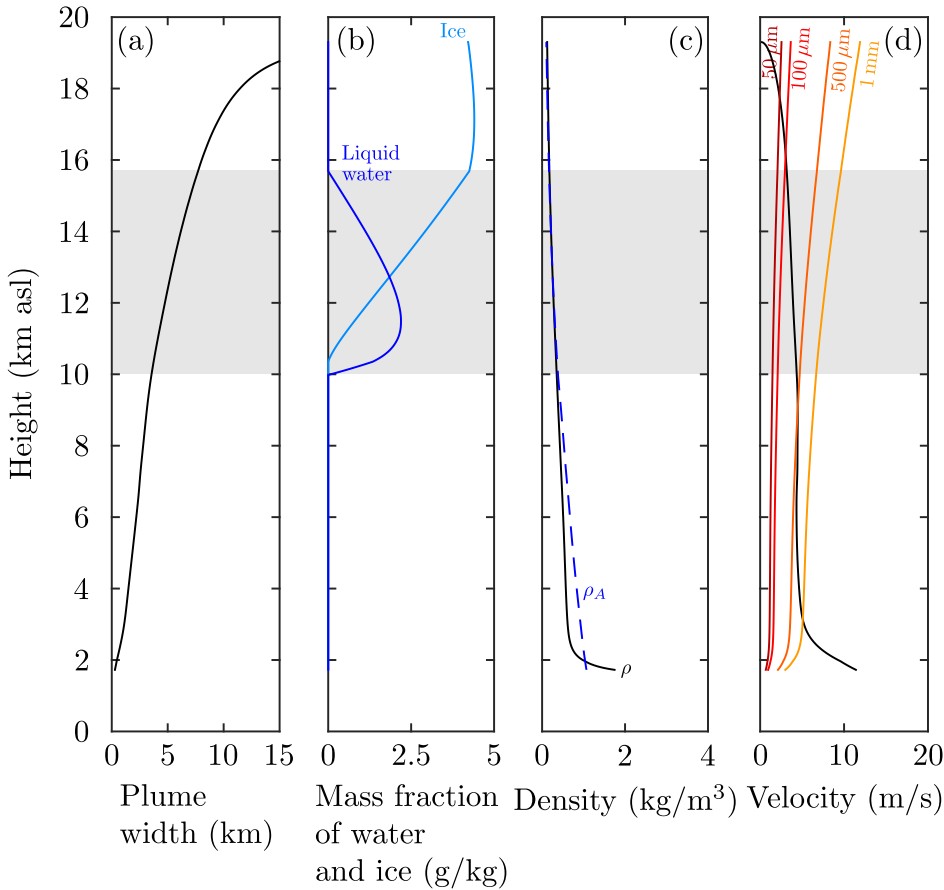

**Figure 13.** Model prediction of the plume at 0500 on 22 May 2011, assuming 10wt% of water vapour at the source. (a) Plume width (taken as twice the Gaussian half-width from the plume centreline) as a function of height. (b) Mass fraction of liquid water and ice as functions of height. (c) Density of the plume $\rho_p$ and atmosphere $\rho_A$ as functions of height. (d) Vertical velocity of the plume at the nominal plume edge (taken as twice the Gaussian half-width from the plume centreline) and critical fall-out velocities of $50\,\mu$m, $100\,\mu$m, $500\,\mu$m and $1\,$mm particles as functions of height.

to aggregation of particles and the formation of mixed phase particles of ice and ash that can lead to rapid fall-out, leaving lighter gases at higher levels in the column. These interactions between the erupting volcanic column and the atmospheric environment in the vicinity of the volcano are important for the short- and long-range transport of gas and particles.

The presence of water in the erupting column, either through additional meltwater or atmospheric entrainment promotes aggregation and facilitates the rapid removal of aggregates from the plume. This lowers the concentrations of ash in the upper parts of the column and may also lead to errors in forecasting ash concentrations in the atmosphere if these processes are not captured in transport models. The photographs shown in Figure 1 and the MODIS satellite image shown in Figure 5 provide strong observational evidence of a lower level 'skirt' of ash moving away from the main ash-rich column.

A combination of satellite observations (passive and active) and dispersion modelling has been used to study the separation of ash and $SO_2$ and it is apparent that such data could be readily utilised in dispersion models by using assimilation (Fu et al., 2017) or through the use of inversion techniques (Stohl et al., 2011). Whether these techniques are sufficiently sensitive to predict separation or perhaps more importantly column collapse and PDC generation, remains to be investigated. The plume model that we use here to analyze the transport of particles in the eruption column highlights the importance of multiphase processes, particularly the role of water in vigorous eruption columns. Clearly more detailed and complex modelling is needed and we recommend that future studies using VATDs consider gases and particles separately and improve parametrisations of the physics of erupting columns. Separation of gases and particles in volcanic eruptions occurs frequently and it seems logical to treat, at least much of the time, the sources separately. Partial column collapse is not an exceptional event and suggests that this process should be included as a mechanism for ash generation and subsequent transport in VATDs. The overwhelming observational evidence for maximum separation with high-level $SO_2$ travelling northwards and low-level ash travelling southwards led to a re-evaluation of model forecasts during the  event, which initially forecast ash collocated with high-level $SO_2$ and covering a large geographic region extending northwards and eastwards from Iceland towards Greenland and the western Norwegian coast (see Fig. 14).

The volcanic ash advisory also shows a region of potentially highly concentrated ash apparently extending to FL200 (20,000 ft or 7 km) travelling southwards and extending eastwards towards Scotland. This erroneous forecast led to closure of airspace over parts of northern Europe and disruption of some air traffic. In the event, observed concentrations did not exceed 2 mg m$^{-3}$ on arrival over northern Europe, and were mostly less than 1 mg m$^{-3}$ Tesche et al. (2012); Ansmann et al. (2012); Moxnes et al. (2014), suggesting that the ash layer was not sufficiently concentrated to be a hazard to aircraft not close to the source; although we caution that without agreed engine manufacturers' tolerance limits the actual dangerous ash concentration (or dosage) is unknown.

Observational data of the kind presented here can be used to constrain VATD models (Stohl et al., 2011) and such models should treat at least the gas and particle components separately. A straightforward way to do this is to extend the methods proposed by Eckhardt et al. (2008), Kristiansen et al. (2010) and Stohl et al. (2011) to include two sources. Improvements in dealing with the complex nature of the interaction of the atmosphere with the erupting column are needed: including better parameterisations of the aggregation process (e.g. Textor et al., 2006; Costa et al., 2010; Telling et al., 2013b; Folch et al., 2016), improved understanding of bent-over plumes (Woodhouse et al., 2013), and improved modelling of the effects of partial

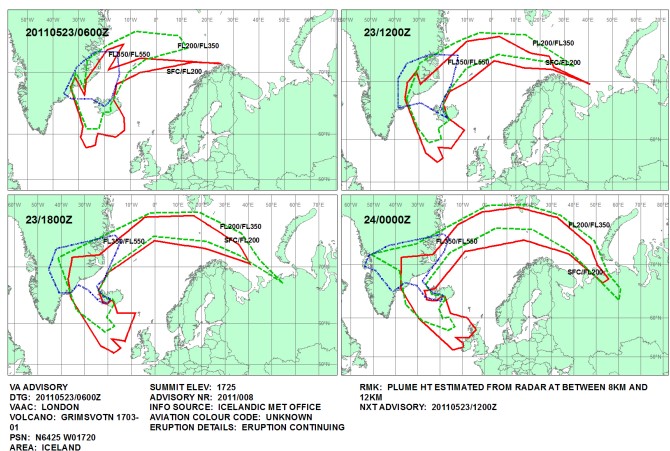

**Figure 14.** Volcanic ash graphic (VAG) issued by the London VAAC on 23 May 2011 at 06:00 UTC. The forecast is valid for the following 18 hours, in 6 hour intervals and shows forecast ash regions from the surface to three flight levels: FL200 (20,000 ft in red), FL350 (35,000 ft, in green dashes) and FL550 (55,000ft, in blue dots).

or total column collapse. While perhaps less common, except in large eruptions (VEI>4), column collapse can lead to the generation of pyroclastic density currents that can act as secondary sources for new column generation, so-called co-ignimbrite plumes (Self and Rampino, 1981). These can be vertically extensive (several kilometres), ash-rich and hence significant in forecasting transport of the aviation hazard. It is suggested here that one or more partial column collapses at  led to surges of 'cold' ash layers that eventually led to transport of ash towards Scotland and southern Scandanavia. This source mechanism is not currently included in ash dispersion models. During the  event the London VAAC used the state-of-the-science dispersion model, NAME (Jones et al., 2007) driven by a source term that relates the total mass erupted to the fourth power of the column height. The fine mass fraction is taken as a small percentage of the total mass; 5% is often used, but this is an unconstrained guess. Clearly, if the column collapses then this parametrisation of the source term is not appropriate.

Emissions of gases and particles into the atmosphere from Icelandic volcanoes can have important consequences for the local environment and also for Europe (Thordarson and Self, 2003). The mechanisms and processes controlling the behaviour of eruptions are vital to understand. Both near-field processes (e.g. ash and gas generation, column collapse, wind structure, aggregation and fall-out) and far-field processes (dispersion, wet/dry deposition, chemical conversion, and aggregation) are important and it is likely that with constraints on these processes better forecasts of the movement of the erupted products can be made.

The transport and fate of $SO_2$ in the atmosphere has implications for the atmospheric radiative balance. Eruptions that generate large amounts of $SO_2$[5] able to penetrate the tropopause can lead to global surface cooling (Robock, 2000) and hence it

---

[5]Stratospheric mass injections of >3 Tg (S) have a measurable impact on the radiative balance

is important to know the vertical emplacement of $SO_2$ from such eruptions. Approximately $\sim$0.13–0.24$\pm$0.1 Tg of $SO_2$ was released by the 21-28 May 2011 eruption of ; nearly all of which resided above the tropopause. The e-folding time for conversion of $SO_2$ to $SO_4^{2-}$ aerosol in the stratosphere is on the order of 20–30 days (Guo et al., 2004), making transport processes likely to cause hemispheric spread of the aerosol. In the case of this eruption, the amount of $SO_2$ released was too small to have a noticeable climate impact. Approximately 0.2–0.4$\pm$0.1 Tg of very fine ash was also released, again too small to have

an appreciable effect on the radiative balance, and not significant enough to cause a hazard to aviation.

*Acknowledgements.* Andrew Hogg and Jeremy Philips provided advice on part of this work and we thank them for their valuable insights. We also thank Antonio Costa and Arnau Folch for providing us with the code to run the FALL3D model

and the NASA AIRS and MODIS science teams for access to the satellite data and products. We acknowledge the use of data products or imagery from the Land, Atmosphere Near real-time Capability for EOS (LANCE) system operated by the NASA/GSFC/Earth Science Data and Information System (ESDIS) with funding provided by NASA/HQ. This work was conducted within the European Commission FUTUREVOLC project. The work of HEH is partially supported by a Leverhulme Emeritus Fellowship. SAC acknowledges support from NASA through grants NNX11AF42G (Aura Science Team)

and NNX13AF50G (MEaSUREs). We thank Arnau Folch and John Stevenson for their reviews of our paper. We are especially grateful to John Stevenson for providing such thorough and thought provoking comments. His comments have helped us improve the manuscript.

**Appendix:  Appendix: Modelling volcanic plumes, aggregation and particle support**

**Model description**

Plume models have been used extensively to examine the dynamics of volcanic eruption columns (Sparks et al., 1997a). While there have been attempts to explicitly model aggregation in plumes (see e.g. Veitch and Woods, 2001; Costa et al., 2010), the models are sensitive to empirical parameters that are not well constrained. Indeed, the physical characteristics (e.g. shape, size, porosity etc.) and chemical composition of volcanic ash particles are likely to greatly alter the aggregation efficiency (James et al., 2002; Durant et al., 2009; Brown et al., 2012; Telling et al., 2013a) and these properties vary substantially for different eruptions. Furthermore, in describing the evolution of aggregating particles, knowledge of the initial particle size distribution is required. The uncertainty introduced by incomplete models, parameters calibrated on small data sets, and unknown initial conditions means that current models of aggregation are unlikely to produce robust predictions for specific events. We instead examine the changing conditions within the plume and assess the effect this could have on the transport of ash particles. This approach does not couple the evolving particle size distribution to the plume dynamics. However, it provides insight into the possibility of rapid aggregation with an abrupt onset.

The plume model of Woodhouse et al. (2013) calculates profiles of plume properties (such as the plume radius, axial velocity, temperature, and the mass fractions of magmatic and atmospheric gases and liquid water) along the plume trajectory, which may be bent-over by the atmospheric wind field. For the  eruption, the wind speeds were not sufficient to significantly affect the plume during its ascent, which was almost vertical. For vertically rising plumes, analogue laboratory experiments (Morton et al., 1956; Papanicolaou and List, 1988) show that the radial profiles of (time-averaged) axial velocity and density deficit are well-described by Gaussian functions. The action of eddies at the margins of the highly turbulent flow in the plume results in entrainment of atmospheric air, which reduces the density difference and eventually (in a stably stratified ambient) the plume reaches the neutral buoyancy height (where the density of the plume equals the atmospheric density) and the plume begins to intrude laterally into the atmosphere (Sparks et al., 1997a; Bursik, 1998; Johnson et al., 2015).

The ash particles transported upwards in the plume are supported by the gaseous phases which exert a drag on the grains sufficient to overcome their weight. Particles can fall out of the plume if they are transported to regions where the gas velocity is not sufficient to support the weight of the grains, which can occur at the plume margins (due to the radial Gaussian profile of vertical velocity) or at a sufficient altitude as the plume decelerates, although fine particle fractions can also be transported into the horizontally intruding layer and subsequently carried to great distances.

The transport and change of phase of water in the plume can play an important role in the plume dynamics (Woods, 1993; Glaze and Baloga, 1996; Woodhouse et al., 2013). Water vapour exsolved from magma or incorporated from surface water or ice around the vent, in addition to water vapour entrained from the moist troposphere, can be carried to high altitude in the relatively hot plume. Cooling of the plume due to entrainment and the reduction in pressure during ascent can result in the plume becoming saturated with respect to water vapour, at which point the water vapour condenses, aided by the presence of condensation nuclei in the form of very fine ash particles (Woods, 1993). The release of latent heat of condensation can lead to a substantial increase in the rise height of the plume in comparison to a dry eruption column that does not become saturated.

This process is particularly important in the moist tropics (Tupper et al., 2009) but can also occur at high latitudes (Woodhouse and Behnke, 2014; Van Eaton et al., 2015). If the plume ascends to altitudes at which the temperature falls below the water freezing temperature, water droplets may begin to freeze. We model ice formation using the approach of Mastin (2007), with a mixture of ice and super-cooled liquid water present for temperatures between $0°C$ and $-40°C$ with mass fractions linearly dependent on the temperature.

To form aggregates, particles must be brought sufficiently close together that electrostatic forces of attraction can bind them or liquid films on the surfaces can coalesce. In the lower region of the plume there is a high concentration of particles, so it might be expected that aggregation occurs here rather than in the upper part of the plume, where entrainment of atmospheric air has greatly reduced the particle concentration. However, the lower part of the plume typically has higher velocities, leading to greater kinetic energy of particle collisions, which reduces the efficiency of aggregation (Telling and Dufek, 2012). The

presence of liquid water increases substantially the aggregation efficiency (Telling et al., 2013a) and, because condensation and freezing typically occur at high altitudes in the plume, the lower velocity of the plume reduces the kinetic energy of collisions. We therefore expect that aggregation proceeds rapidly in 'wet' conditions, resulting in a pronounced increase in the size of particle clusters, while electrostatically dominated aggregation in 'dry' regions results in more gradual growth of clusters.

We consider a particle of diameter $d$ and density $\rho_s$ that is transported with speed $u_s$ in the plume that is rising with vertical velocity $u_p$. The hydrodynamic drag acting on the particle is given by

$$F_D = \frac{\pi d^2}{8} \rho_p C_D \left(u_s - u_p\right)^2 \mathrm{sgn}\left(u_p - u_s\right),$$  (1)

where $\rho_p$ is the bulk density of the plume and $C_D$ is the drag coefficient of the particle. Balancing drag with the weight of the particle at the point when the particle is no longer supported by the plume (i.e. $u_s = 0$), we find that

$$\frac{\pi d^2}{8} \rho_p C_D u_p^2 = \frac{\pi d^3}{6} \rho_s g$$  (2)

and therefore the particle falls out of the plume when

$$u_p \leqslant \left(\frac{4 d \rho_s g}{3 \rho_p C_D}\right)^{1/2} \equiv u_c(d),$$  (3)

where $u_c(d)$ is the critical fall-out velocity for a particle of diameter $d$. The value of the drag coefficient depends on properties of the particle, particularly shape, and on the Reynolds number of the flow field in which it is carried (Wilson and Huang,

1979). Furthermore, the drag coefficient of aggregates may differ from that of individual particles (James et al., 2003). Here we take a representative value of $C_D = 1$ noting that $u_c(d)$ is not strongly sensitive to the value of the drag coefficient. The variation in density of solids (from $\sim 700$ kg m$^{-3}$ for vesicular pumice to $\sim 3200$ kg m$^{-3}$ for glass shards) does not greatly alter the critical fall-out velocity calculated using our reference density (1200 kg m$^{-3}$) with changes in the value by a factor of 0.76 to 1.6.

We assume that the radial profile of the mean axial velocity of the plume is Gaussian,

$$u_p(r, z) = u(z) \exp\left(-r^2/R^2\right),$$  (4)

where $r$ is the radial distance from the centreline of the plume and $R$ is a characteristic radial length scale. The radial distance $r = 2R$ is taken as representative of the plume width, and at that point the local mean axial velocity of the plume is less than $2\%$ of the centreline value.

**Model Results**

At 05:00 UTC on 22 May 2011, the C-band weather radar at Keflavik International Airport recorded a plume height of $19.3\,\text{km}$. The mass flux of erupted material is estimated by matching the model prediction of the plume height to the radar observation with fixed values of the vent radius, gas mass fraction and temperature at the source (Table 3). The resulting source mass flux estimate is $Q_0 = 9.5 \times 10^7$ kg s$^{-1}$.

**Table 3.** Source parameter value used in the plume model for 0500 on 22 May 201.

| Parameter (symbol) | Value |
| --- | --- |
| Vent radius ($L_0$) | $200\,\text{m}$ |
| Source gas mass fraction ($n_0$) | 0.05 |
| Source temperature ($T_0$) | $1000\,\text{K}$ |
| Vent altitude ($z_0$) | $1725\,\text{m}$ |

Figure 15 shows time series of the plume height and condensation level, the maximum mass fractions of liquid water and ice in the plume, and the critical height at which particles fall out of the plume for four particle diameters on the 22 May 2011. Plume top heights are derived from a fixed C-band radar and a mobile X-band radar. The variation of the condensation level in the plume follows that of the plume top height. The mass fractions of liquid water and ice in the plume do not vary substantially (with the exception of a decrease in the ice content at 09:00 and 10:00 UTC) despite changes in the condensation height, and there is a plentiful supply of condensed water in the plume throughout this period of the eruption. There are pronounced differences in the critical fall-out heights of particles of different diameters. Particles of diameter $50\,\mu\text{m}$ are carried to near to the plume top height, above the condensation level. Often $100\,\mu\text{m}$ diameter particles are carried above the condensation level, but we note that between 09:00 and 10:00 UTC on 22 May the critical fall-out velocity of these particles is reached at low levels in the plume. The larger particles (diameters of $500\,\mu\text{m}$ and $1\,\text{mm}$) consistently fall out below the condensation level. The period between 09:00 and 10:00 UTC on 22 May is distinctive in the relatively low plume height, ice content and low critical fall-out height for particles of diameter greater than $500\,\mu\text{m}$. The low plume height requires a reduced mass flux from the source and therefore relatively low velocities in the plume. Thus the critical fall-out velocity of a particle occurs at lower altitudes. Therefore, during this period the fall out of relatively small diameter particles could occur without significant wet aggregation; dry aggregation in the lower plume might be sufficient to remove very fine ash.

The model source conditions used above (Table 3) have a relatively dry source with a water vapour mass fraction of $5\,\text{wt}\%$. However, the melting of glacier ice around the vent at  is likely to have contributed water vapour in addition to that derived

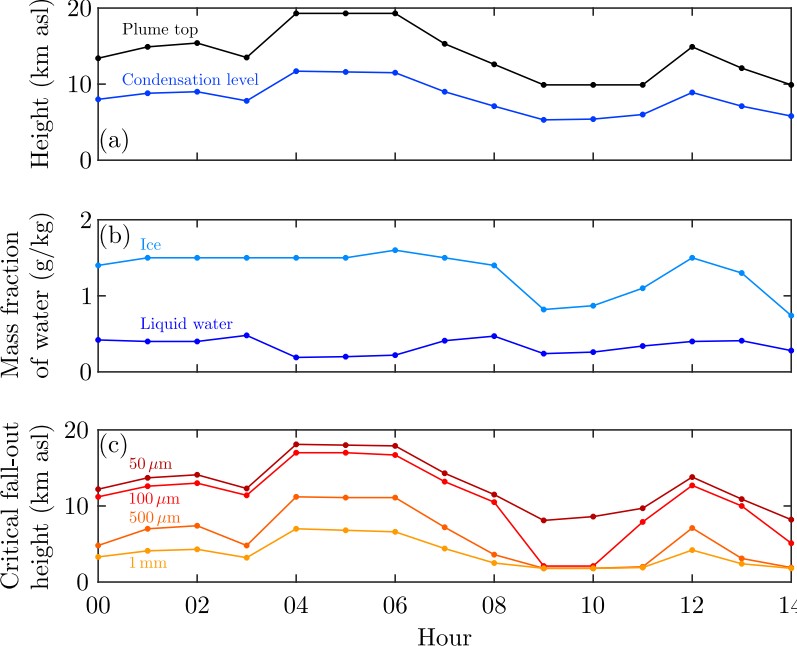

**Figure 15.** Model predictions of the properties of the  plume on 22 May 2011. (a) Plume top height and condensation level in the plume. (b) Maximum mass fractions of liquid water and water ice. (c) Critical height at which particles fall-out of the plume for particles of diameters $50\,\mu$m, $100\,\mu$m, $500\,\mu$m and $1\,$mm.

from magma. The sensitivity of the model predictions to the source water mass fraction is examined in Figure 16 where the water vapour content is taken to be $5\,$wt%, $10\,$wt% and $15\,$wt%. Adding water at the source has a pronounced effect on the condensed water content of the plume, with both the mass fractions of the condensed phases increasing and the level at which condensation occurs decreasing as the source mass fraction of water vapour increases. When the source is relatively dry, with $n_0 = 0.05$, condensation occurs when the plume temperature is below $0°$C, so liquid water and ice are expected to form. In contrast, for both $n_0 = 0.1$ and $n_0 = 0.15$ the condensation occurs when the plume temperature exceeds $0°$C and therefore the vapour first condenses to water, with ice forming at higher altitudes as the temperature decreases. We note that the source mass fraction of water vapour strongly influences the buoyancy of the erupted material at the source; for $n_0 = 0.05$ and $n_0 = 0.1$ the erupted material is initially more dense than the atmosphere and is driven upwards by momentum, whereas the material is buoyant at the vent when $n_0 = 0.15$. Interestingly, the velocity at the source when $n_0 = 0.15$ is greater than that when $n_0 = 0.1$. However, the dependence of the fall-out velocity on the plume density means that the fall-out height for each particle size decreases substantially as $n_0$ increases.

Figure 16 demonstrates that the potential for the separation of very fine ash from the plume, driven by wet aggregation, increases substantially as the source water vapour content increases. However, for the atmospheric conditions at the time of the  eruption, the model predicts substantial concentrations of condensed water for all of the source conditions examined.

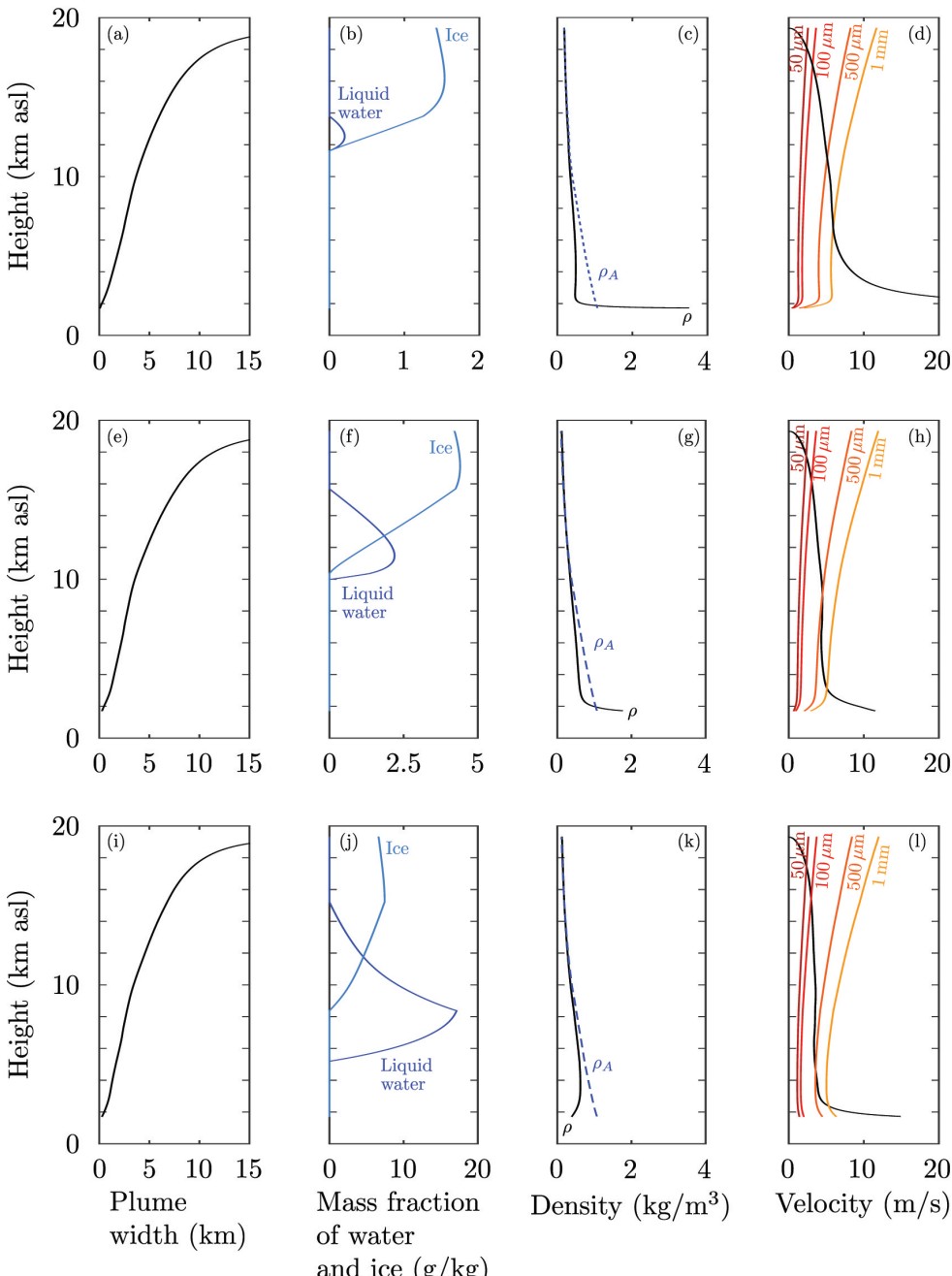

**Figure 16.** Sensitivity of model predictions of the plume at 0500 on 22 May 2011 to increases in the source water vapour content, with (a-d) $n_0 = 0.05$, (e-h) $n_0 = 0.10$ and (i-l) $n_0 = 0.15$. (a,e,i) Plume width as a function of height. (b,f,j) Mass fraction of liquid water and water ice as a function of height. (c,g,k) Density of the plume $\rho_p$ and atmosphere $\rho_A$ as functions of height. (d,h,l) Vertical velocity of the plume at the plume edge and critical fall-out velocities of $50\,\mu$m, $100\,\mu$m, $500\,\mu$m and $1\,$mm particles as functions of height.

Therefore, our hypothesis of water-mediated aggregation and enhanced removal of ash from the plume is robust to changes in the source conditions.

## Appendix:  Satellite data retrievals

A sequence of FALL3D ash simulations and satellite-based retrievals of $SO_2$ gas and very fine ash are included as Supplementary material.

## Appendix:  Photographs

Photographs of the developing  eruption column are provided with some annotations are also provided as Supplementary Material (Photographs). These show the complex nature of the early phase of the eruption and possible partial column collapse and initiation of a PDC.

## 10 Appendix: References

Ansmann, A., Seifert, P., Tesche, M., and Wandinger, U.: Profiling of fine and coarse particle mass: case studies of Saharan dust and Eyjafjallajökull/Grimsvötn volcanic plumes, Atmospheric Chemistry and Physics, 12, 9399–9415, 2012.

Arason, P., Petersen, G., and Bjornsson, H.: Observations of the altitude of the volcanic plume during the eruption of Eyjafjallajökull, April–May 2010, Earth System Science Data Discussions, 4, 1–25, 2011.

Bourassa, A. E., Robock, A., Randel, W. J., Deshler, T., Rieger, L. A., Lloyd, N. D., Llewellyn, E. T., and Degenstein, D. A.: Large volcanic aerosol load in the stratosphere linked to Asian monsoon transport, Science, 337, 78–81, 2012.

Brown, R., Bonadonna, C., and Durant, A.: A review of volcanic ash aggregation, Physics and Chemistry of the Earth, 45–46, 65–78, doi:doi:10.1016/j.pce.2011.11.001, 2012.

Bursik, M.: Tephra dispersal, Geological Society, London, Special Publications, 145, 115–144, doi:10.1144/GSL.SP.1996.145.01.07, 1998.

Bursik, M.: Effect of wind on the rise height of volcanic plumes, Geophysical Research Letters, 28, 3621–3624, 2001.

Carey, S. and Bursik, M.: Volcanic plumes, Encyclopedia of volcanoes. Academic Press, San Diego, pp. 572–585, 2015.

Carn, S., Strow, L., de Souza-Machado, S., Edmonds, Y., and Hannon, S.: Quantifying tropospheric volcanic emissions with AIRS: The 2002 eruption of Mt. Etna (Italy), Geophysical Research Letters, 32, 2005.

Carn, S., Clarisse, L., and Prata, A.: Multi-decadal satellite measurements of global volcanic degassing, Journal of Volcanology and Geother-
mal Research, 2016.

Clarisse, L. and Prata, F.: Infrared sounding of volcanic ash *In Volcanic Ash, Ed. S. Mackie*, Elsevier, 2016.

Clarisse, L., Coheur, P.-F., Prata, A. J., Hurtmans, D., Razavi, A., Phulpin, T., Hadji-Lazaro, J., and Clerbaux, C.: Tracking and quantifying volcanic $SO_2$ with IASI, the September 2007 eruption at Jebel at Tair, Atmospheric chemistry and physics, 8, 7723–7734, 2008.

Clarisse, L., Prata, F., Lacour, J.-L., Hurtmans, D., Clerbaux, C., and Coheur, P.-F.: A correlation method for volcanic ash detection using
hyperspectral infrared measurements, Geophysical research letters, 37, 2010.

Costa, A., Folch, A., and Macedonio, G.: A model for wet aggregation of ash particles in volcanic plumes and clouds: 1. Theoretical formulation, Journal of Geophysical Research: Solid Earth (1978–2012), 115, 2010.

Degruyter, W. and Bonadonna, C.: Improving on mass flow rate estimates of volcanic eruptions, Geophysical Research Letters, 39, 2012.

Draxler, R. and Rolph, G.: HYSPLIT (HYbrid Single-Particle Lagrangian Integrated Trajectory) model access via NOAA ARL READY
website (http://www. arl. noaa. gov/ready/hysplit4. html). NOAA Air Resources Laboratory, Silver Spring, 2003.

Durant, A., Rose, W., Sarna-Wojcicki, A., Carey, S., and Volentik, A.: Hydrometeor-enhanced tephra sedimentation: Constraints from the 18 May 1980 eruption of Mount St. Helens, Journal of Geophysical Research, 114, B03 204, doi:10.1029/2008JB005756, 2009.

Eckhardt, S., Prata, A., Seibert, P., Stebel, K., and Stohl, A.: Estimation of the vertical profile of sulfur dioxide injection into the atmosphere by a volcanic eruption using satellite column measurements and inverse transport modeling, Atmospheric Chemistry and Physics, 8,
3881–3897, 2008.

Folch, A., Costa, A., and Macedonio, G.: FALL3D: A computational model for transport and deposition of volcanic ash, Computers & Geosciences, 35, 1334–1342, 2009.

Folch, A., Costa, A., and Macedonio, G.: FPLUME-1.0: An integral volcanic plume model accounting for ash aggregation, Geoscientific Model Development, 9, 431, 2016.

Fromm, M., Kablick, G., Nedoluha, G., Carboni, E., Grainger, R., Campbell, J., and Lewis, J.: Correcting the record of volcanic stratospheric aerosol impact: Nabro and Sarychev Peak, Journal of Geophysical Research: Atmospheres, 119, 2014.

Fu, G., Prata, A., Lin, H., Heemink, A., Segers, A., and Lu, S.: Data assimilation for volcanic ash plumes using a satellite observational operator: a case study on the 2010 Eyjafjallajökull volcanic eruption, Atmos. Chem. Phys., 17, 1187–1205, 2017.

Glaze, L. and Baloga, S.: Sensitivity of buoyant plume heights to ambient atmospheric conditions: Implications for volcanic eruption columns, Journal of Geophysical Research, 101, 1529–1540, doi:10.1029/95JD03071, 1996.

Gudmundsson, M., Höskuldsson, Á., Larsen, G., Thordarson, T., Oladottir, B., Oddsson, B., Gudnason, J., Högnadottir, T., Stevenson, J., Houghton, B., et al.: The May 2011 eruption of Grímsvötn, in: EGU General Assembly Conference Abstracts, vol. 14, p. 12119, 2012a.

Gudmundsson, M. T., Thordarson, T., Höskuldsson, Á., Larsen, G., Björnsson, H., Prata, F. J., Oddsson, B., Magnússon, E., Högnadóttir, T., Petersen, G. N., et al.: Ash generation and distribution from the April-May 2010 eruption of Eyjafjallajökull, Iceland, Scientific reports, 2, 2012b.

Guo, S., Bluth, G. J., Rose, W. I., Watson, I. M., and Prata, A.: Re-evaluation of SO2 release of the 15 June 1991 Pinatubo eruption using ultraviolet and infrared satellite sensors, Geochemistry, Geophysics, Geosystems, 5, 2004.

Hoffmann, L., Grießbach, S., and Meyer, C. I.: Volcanic emissions from AIRS observations: detection methods, case study, and statistical analysis, in: Proceedings of the SPIE, p. 924214, Forschungszentrum Jülich GmbH (Germany), 2014.

Holasek, R. E., Woods, A. W., and Self, S.: Experiments on gas-ash separation processes in volcanic umbrella plumes, Journal of volcanology and geothermal research, 70, 169–181, 1996.

Hunt, W. H., Winker, D. M., Vaughan, M. A., Powell, K. A., Lucker, P. L., and Weimer, C.: CALIPSO Lidar Description and Performance Assessment, J. Atmos. Ocean. Tech., 26, 1214–1228, 2009.

James, M., Gilbert, J., and Lane, S.: Experimental investigation of volcanic particle aggregation in the absence of liquid phase, Journal of Geophysical Research, 107, 2191, doi:10.1029/2001JB000950, 2002.

James, M., Lane, S., and Gilbert, J.: Density, construction, and drag coefficient of electrostatic volcanic ash aggregates, Journal of Geophysical Research, 108, 2435, doi:10.1029/2002JB002011, 2003.

Johnson, C., Hogg, A., Huppert, H., Sparks, R., Phillips, J., Slim, A., and Woodhouse, M.: Modelling Intrusions through quiescent and moving ambients, Journal of Fluid Mechanics, 771, 370 – 406, doi:10.1017/jfm.2015.180, 2015.

Jones, A., Thomson, D., Hort, M., and Devenish, B.: The UK Met Office's next-generation atmospheric dispersion model, NAME III, in: Air Pollution Modeling and its Application XVII, pp. 580–589, Springer, 2007.

Jude-Eton, T., Thordarson, T., Gudmundsson, M., and Oddsson, B.: Dynamics, stratigraphy and proximal dispersal of supraglacial tephra during the ice-confined 2004 eruption at Grímsvötn Volcano, Iceland, Bulletin of volcanology, 74, 1057–1082, 2012.

Kristiansen, N., Stohl, A., Prata, A., Richter, A., Eckhardt, S., Seibert, P., Hoffmann, A., Ritter, C., Bitar, L., Duck, T., et al.: Remote sensing and inverse transport modeling of the Kasatochi eruption sulfur dioxide cloud, Journal of Geophysical Research: Atmospheres (1984–2012), 115, 2010.

Kylling, A., Kahnert, M., Lindqvist, H., and Nousiainen, T.: Volcanic ash infrared signature: porous non-spherical ash particle shapes compared to homogeneous spherical ash particles, Atmospheric Measurement Techniques, 7, 919–929, 2014.

Mastin, L.: A user-friendly one-dimensional model for wet volcanic plumes, Geochemistry, Geophysics, Geosystems, 8, Q03 014, doi:10.1029/2006GC001455, 2007.

Mastin, L., Guffanti, M., Servranckx, R., Webley, P., Barsotti, S., Dean, K., Durant, A., Ewert, J., Neri, A., Rose, W., et al.: A multidisciplinary effort to assign realistic source parameters to models of volcanic ash-cloud transport and dispersion during eruptions, Journal of Volcanology and Geothermal Research, 186, 10–21, 2009.

Morton, B., Taylor, G., and Turner, J.: Turbulent Gravitational Convection from Maintained and Instantaneous Sources, Proceedings of the Royal Society of London. Series A. Mathematical and Physical Sciences, 234, 1–23, doi:10.1098/rspa.1956.0011, 1956.

Moxnes, E. D., Kristiansen, N. I., Stohl, A., Clarisse, L., Durant, A., Weber, K., and Vogel, A.: Separation of ash and sulfur dioxide during the 2011 Grímsvötn eruption, Journal of Geophysical Research: Atmospheres, 119, 7477–7501, 2014.

Papanicolaou, P. and List, E.: Investigations of round vertical turbulent buoyant jets., Journal of Fluid Mechanics, 195, 341–391, doi:10.1017/S0022112088002447, 1988.

Petersen, G., Bjornsson, H., Arason, P., and Löwis, S. v.: Two weather radar time series of the altitude of the volcanic plume during the May 2011 eruption of Grímsvötn, Iceland, Earth System Science Data Discussions, 5, 281–299, 2012.

Prata, A.: Satellite detection of hazardous volcanic clouds and the risk to global air traffic, Natural hazards, 51, 303–324, 2009.

Prata, A. and Bernardo, C.: Retrieval of volcanic $SO_2$ column abundance from Atmospheric Infrared Sounder data, Journal of Geophysical Research: Atmospheres (1984–2012), 112, 2007.

Prata, A. and Grant, I.: Retrieval of microphysical and morphological properties of volcanic ash plumes from satellite data: Application to Mt Ruapehu, New Zealand, Quarterly Journal of the Royal Meteorological Society, 127, 2153–2179, 2001.

Prata, A. and Prata, A.: Eyjafjallajökull volcanic ash concentrations determined using Spin Enhanced Visible and Infrared Imager measurements, Journal of Geophysical Research: Atmospheres (1984–2012), 117, 2012.

Prata, A. and Rose, W.: Volcanic ash hazards to aviation, Encyclopedia of Volcanoes. Chapter 52, 2015.

Prata, A. J., Gangale, G., Clarisse, L., and Karagulian, F.: Ash and sulfur dioxide in the 2008 eruptions of Okmok and Kasatochi: Insights from high spectral resolution satellite measurements, Journal of Geophysical Research: Atmospheres, 115, n/a–n/a, doi:10.1029/2009JD013556, 2010.

Prata, A. T., Siems, S. T., and Manton, M. J.: Quantification of volcanic cloud top heights and thicknesses using A-train observations for the 2008 Chaitén eruption, J. Geophys. Res. Atmos., 120, 2928–2950, 2015.

Prata, A. T., Young, S. A., Siems, S. T., and Manton, M. J.: Lidar ratios of stratospheric volcanic ash and sulfate aerosols retrieved from CALIOP measurements, Atmospheric Chemistry and Physics Discussions, 2017, 1–28, doi:10.5194/acp-2016-1173, http://www.atmos-chem-phys-discuss.net/acp-2016-1173/, 2017.

Rienecker, M. M., Suarez, M. J., Todling, R., Bacmeister, J., Takacs, L., Liu, H. C., Gu, W., Sienkiewicz, M., Koster, R. D., Gelaro, R., Stajner, I., and Nielsen, J. E.: The GEOS-5 Data Assimilation System— Documentation of Versions 5.0.1, 5.1.0, and 5.2.0, Technical Report Series on Global Modeling and Data Assimilation, 27, NASA/TM-2008-104606, 1–118, 2008.

Robock, A.: Volcanic eruptions and climate, Reviews of Geophysics, 38, 191–219, 2000.

Self, S. and Rampino, M. R.: The 1883 eruption of Krakatau, Nature, 294, 699–704, 1981.

Sigmarsson, O., Haddadi, B., Carn, S., Moune, S., Gudnason, J., Yang, K., and Clarisse, L.: The sulfur budget of the 2011 Grímsvötn eruption, Iceland, Geophysical Research Letters, 40, 6095–6100, 2013.

Sparks, R.: The dimensions and dynamics of volcanic eruption columns, Bulletin of Volcanology, 48, 3–15, doi:10.1007/BF01073509, 1986.

Sparks, R., Bursik, M., Carey, S., Gilbert, J., Glaze, L., Sigurdsson, H., and Woods, A.: Volcanic Plumes, John Wileys & Sons, 1997a.

Sparks, R. S. J., Bursik, M., Carey, S., Gilbert, J., Glaze, L., Sigurdsson, H., and Woods, A.: Volcanic plumes, Wiley, 1997b.

Stevenson, J., Millington, S., Beckett, F., Swindles, G., and Thordarson, T.: Big grains go far: reconciling tephrochronology with atmospheric measurements of volcanic ash., Atmospheric Measurement Techniques Discussions, 8, 2015.

Stevenson, J. A., Loughlin, S., Rae, C., Thordarson, T., Milodowski, A. E., Gilbert, J. S., Harangi, S., Lukács, R., Höjgaard, B., Árting, U., Pyne-O'Donnell, S., MacLeod, A., Whitney, B., and Cassidy, M.: Distal deposition of tephra from the Eyjafjallajökull 2010 summit eruption, Journal of Geophysical Research: Solid Earth, 117, n/a–n/a, doi:10.1029/2011JB008904, 2012.

Stevenson, J. A., Loughlin, S. C., Font, A., Fuller, G. W., MacLeod, A., Oliver, I. W., Jackson, B., Horwell, C. J., Thordarson, T., and Dawson, I.: UK monitoring and deposition of tephra from the May 2011 eruption of Grímsvötn, Iceland, Journal of Applied Volcanology, 5     2, 3, 2013.

Stohl, A., Prata, A. J., Eckhardt, S., Clarisse, L., Durant, A., Henne, S., Kristiansen, N. I., Minikin, A., Schumann, U., Seibert, P., Stebel, K., Thomas, H. E., Thorsteinsson, T., Tørseth, K., and Weinzierl, B.: Determination of time- and height-resolved volcanic ash emissions and their use for quantitative ash dispersion modeling: the 2010 Eyjafjallajökull eruption, Atmospheric Chemistry and Physics, 11, 4333–4351, doi:10.5194/acp-11-4333-2011, 2011.

Telling, J. and Dufek, J.: An experimental evaluation of ash aggregation in explosive volcanic eruptions, Journal of Volcanology and Geothermal Research, 209–210, 1–8, doi:10.1016/j.jvolgeores.2011.09.008, 2012.

Telling, J., Dufek, J., and Shaikh, A.: Ash aggregation in explosive volcanic eruptions, Geophysical Research Letters, 40, 2355–2360, doi:10.1002/grl.50376, 2013a.

Telling, J., Dufek, J., and Shaikh, A.: Ash aggregation in explosive volcanic eruptions, Geophys. Res. Lett., 40, 2355–2360, 2013b.

Tesche, M., Glantz, P., Johansson, C., Norman, M., Hiebsch, A., Ansmann, A., Althausen, D., Engelmann, R., and Seifert, P.: Volcanic ash over Scandinavia originating from the Grímsvötn eruptions in May 2011, Journal of Geophysical Research: Atmospheres, 117, 2012.

Textor, C., Graf, H.-F., Herzog, M., Oberhuber, J. M., Rose, W. I., and Ernst, G. G.: Volcanic particle aggregation in explosive eruption columns. Part I: Parameterization of the microphysics of hydrometeors and ash, Journal of volcanology and geothermal research, 150, 359–377, 2006.

Thomas, H. E. and Prata, A. J.: Sulphur dioxide as a volcanic ash proxy during the April and May 2010 eruption of Eyjafjallajökull Volcano, Iceland, Atmospheric Chemistry and Physics, 11, 6871–6880, doi:10.5194/acp-11-6871-2011, 2011.

Thordarson, T. and Larsen, G.: Volcanism in Iceland in historical time: Volcano types, eruption styles and eruptive history, Journal of Geodynamics, 43, 118–152, 2007.

Thordarson, T. and Self, S.: Atmospheric and environmental effects of the 1783–1784 Laki eruption: a review and reassessment, Journal of 25    Geophysical Research, 108, 4011, 2003.

Tupper, A., Textor, C., Herzog, M., Graf, H.-F., and Richards, M. S.: Tall clouds from small eruptions: the sensitivity of eruption height and fine ash content to tropospheric instability, Natural Hazards, 51, 375–401, doi:10.1007/s11069-009-9433-9, 2009.

Van Eaton, A., Muirhead, J., Wilson, C., and Cimarelli, C.: Growth of volcanic ash aggregates in the presence of liquid water and ice: an experimental approach, Bulletin of Volcanology, 74, 1963–1984, doi:10.1007/s00445-012-0634-9, 2012.

Van Eaton, A., Mastin, L., Herzog, M., Schwaiger, H., Schneider, D., Wallace, K., and Clarke, A.: Hail formation triggers rapid ash aggregation in volcanic plumes, Nature Communications, 6, 1–7, doi:10.1038/ncomms8860, 2015.

Vaughan, M. A., Powell, K. A., Kuehn, R. E., Young, S. A., Winker, D. M., Hostetler, C. A., Hunt, W. H., Liu, Z., McGill, M. J., and Getzewich, B. J.: Fully Automated Detection of Cloud and Aerosol Layers in the CALIPSO Lidar Measurements, J. Atmos. Ocean. Tech., 26, 2034–2050, 2009.

Veitch, G. and Woods, A.: Particle aggregation in volcanic eruption columns, Journal of Geophysical Research: Solid Earth, 106, 26 425–26 441, doi:10.1029/2000JB900343, 2001.

Wen, S. and Rose, W. I.: Retrieval of sizes and total masses of particles in volcanic clouds using AVHRR bands 4 and 5, Journal of Geophysical Research: Atmospheres (1984–2012), 99, 5421–5431, 1994.

Wilson, L. and Huang, T.: The influence of shape on the atmospheric settling velocity of volcanic ash particles, Earth and Planetary Science Letters, 44, 311–324, doi:10.1016/0012-821X(79)90179-1, 1979.

Wilson, L., Sparks, R., Huang, T., and Watkins, N.: The Control of Volcanic Column Heights by Eruption Energetics and Dynamics, Journal of Geophysical Research, 83, 1829–1836, 1978.

Winker, D. M., Liu, Z., Omar, A., Tackett, J., and Fairlie, D.: CALIOP observations of the transport of ash from the Eyjafjallajökull volcano in April 2010, J. Geophys. Res. Atmos., 117, D00U15, 2012.

Woodhouse, M. and Behnke, S.: Charge structure in volcanic plumes: a comparison of plume properties predicted by an integral plume model to observations of volcanic lightning during the 2010 eruption of Eyjafjallajökull, Iceland, Bulletin of Volcanology, 76, doi:10.1007/s00445-014-0828-4, 2014.

Woodhouse, M., Hogg, A., Phillips, J., and Sparks, R.: Interaction between volcanic plumes and wind during the 2010 Eyjafjallajökull eruption, Iceland, Journal of Geophysical Research, 118, doi:10.1029/2012JB009592., 2013.

Woods, A.: The fluid dynamics and thermodynamics of eruption columns, Bulletin of Volcanology, 50, 169–193, doi:The fluid dynamics and thermodynamics of eruption columns, 1988.

Woods, A.: Moist Convection and the Injection of Volcanic Ash Into the Atmosphere, Journal of Geophysical Research, 98, 17 627–17 636, doi:10.1029/93JB00718, 1993.

Woods, A. W., Holasek, R. E., and Self, S.: Wind-driven dispersal of volcanic ash plumes and its control on the thermal structure of the plume-top, Bulletin of volcanology, 57, 283–292, 1995.

Yang, K., Krotkov, N. A., Krueger, A. J., Carn, S. A., Bhartia, P. K., and Levelt, P. F.: Retrieval of large volcanic $SO_2$ columns from the Aura Ozone Monitoring Instrument: Comparison and limitations, Journal of Geophysical Research: Atmospheres (1984–2012), 112, 2007a.

Yang, P., Feng, Q., Hong, G., Kattawar, G. W., Wiscombe, W. J., Mishchenko, M. I., Dubovik, O., Laszlo, I., and Sokolik, I. N.: Modeling of the scattering and radiative properties of nonspherical dust-like aerosols, Journal of Aerosol Science, 38, 995 – 1014, doi:https://doi.org/10.1016/j.jaerosci.2007.07.001, http://www.sciencedirect.com/science/article/pii/S0021850207001061, 2007b.