# Peer review of "Atmospheric processes affecting the separation of volcanic ash and $SO_2$ in volcanic eruptions"

_Atmospheric Chemistry and Physics, 2017_

## Referee Comment (RC1) · J.A. Stevenson (Referee) · 4 Mar 2017

**1   Summary**

The paper reports on the dispersal of volcanic ash and sulphur dioxide from the May 2011 eruption of Grímsvötn volcano, Iceland. It demonstrates that most of the the ash, which was mainly present in the lowest 4 km of the plume, was transported southwards and then east to Europe; the sulphur dioxide travelled northwards at higher altitudes. Various satellite data were used. Infrared and LiDAR measured the altitude of the ash cloud.  Ash concentration and grainsize were retrieved from infrared data.  Infrared and ultraviolet data were used to retrieve concentration and erupted mass of SO2.

Dispersion modelling found that smaller mass eruption rates than would be calculated based on the plume height gave a better match to satellite observations. A model of plume physics demonstrates that addition of meltwater resulted in a large region where liquid water was present. This could promote aggregation and premature deposition of fine ash. The study concludes that forecasts of ash concentration and our general understanding of volcanic cloud behaviour would be improved if volcanic ash and SO2 were modelled as independent components.

**2  Contribution**

Understanding the separation of volcanic ash from sulphur dioxide is important to the wider understanding of volcanic plume processes and the hazards presented by air-borne volcanic ash. Such separation is most common in lava-producing eruptions. This paper provides a striking example of separation during an explosive eruption where much extremely fine ash was produced. As such, it is a very interesting contribution.

The satellite methods used are all well-established, as was the tool used for modelling dispersion of volcanic ash. The modelling of aggregation within the plume is a new application of an existing volcanic plume model.

The main conclusion, that most of the volcanic ash was transported in a different direction from the sulphur dioxide gas is well supported by the data. However, there are other statements that are not well supported and should be dropped. These include:

- "separation of material ... is inevitable ... in presence of wind shear"

- that "column collapse" was the reason for the low level of the ash-rich plume

- that the ash cloud "was not significant enough to cause a hazard to aviation"

Instead, I think that the following conclusions deserve more emphasis.

- Modelling of plume physics shows that addition of meltwater leads to a large region in the plume where water is present in the liquid phase. This promotes aggregation of ash and premature deposition of finer particles, leading to low concentrations of ash in the upper part of the plume.

- The separation of gas and ash was a consequence of *both* aggregation *and* wind shear.

- The Mastin equation, which relates discharge rate to the fourth power of the plume height and which is based on data from 'dry' plumes, is not appropriate when eruptions take place through water or ice.

- Satellite data are very useful in evaluating dispersion model predictions and this eruption highlights a situation where refining dispersion model source parameters using inversion techniques could have resulted in improved predictions.

**3 Suitability for publication**

The main findings of the paper represent an important case study and should be published. However, there are areas where major revisions are required before the paper is suitable for publication.
**4   Areas for revision**

**4.1   Missing reference**

The authors should discuss their results in the context of findings of Stevenson et al (2013), who measured deposition in the UK of ash from the Grímsvötn 2011 eruption. The most relevant findings were that:

- The majority of ash deposition was in Scotland on 24-25 May (in agreement with modelling results and satellite observations presented here).

- Air mass trajectories showed that material from the lowest 4 km of the plume was transported southwards, while material higher up was transported northwords (in agreement with modelling results and satellite observations presented here).

- Ash deposited in Scotland had a median grainsize of 19-23 microns and a maximum of 80 microns (significantly higher than suggested by the effective radius values presented here).

- Pollen traps in Scotland collected tiny (3-5 micron length) ash particles that fell within rain drops on 27 May. Trajectory analysis showed that these came from the upper part of the plume and had travelled via Greenland. Thus, there was at least some ash present in the material that went north. (Raising the question of why it was not detected in the IR retrievals or why the particles appear spherical to Caliop LiDAR).

**4.2   Discussion of physical volcanology and plume 'collapse'**

The discussion of physical volcanology is speculative and contains some errors. The figure shows Grímsvötn as a conical peak, when the eruption took place in a caldera

lake, and shows the SO2 and ash clouds moving in the same direction. The text uses out-dated terminology (e.g. pyroclastic flows instead of pyroclastic density currents) and contains factually incorrect statements e.g. that the meltwater 'added' energy to the plume (it absorbs it from the magma during vaporisation and releases it again at altitude when it condenses).

I think that this section should be trimmed by removing the discussions of collapse mechanism, which are not informed by new data from this study. The focus on aggregates, which are explained by the subsequent modelling section, should be sharpened.

Nevertheless, I also recommend that the authors cite Jude Eton et al (2012), which describes the plume and the deposits from the 2004 eruption at Grímsvötn. I have worked on the 2011 deposits in the field (as has Thordarson) and they are very similar and indicate that the same processes took place. The paper contains very instructive photographs of ground-hugging, laterally-emplaced density currents that were sourced from the eruption column that do not require 'collapse' of the plume.

4.3   Uncertainty estimates for ash retrievals (Figure 7)

On Page 16, the authors state that "the error in estimating fine ash mass from infrared retrievals has been investigated by Wen and Rose (1994) and Prata and Prata (2012), who suggest errors of 40-50%". I would suggest that this is strongly underestimated.

There are two lines of evidence within this study that suggest this.

Firstly, the authors state repeatedly that much of the ash cloud is covered by meteorological cloud. This will prevent detection in those regions and is a large source of uncertainty. Therefore, any estimates should be treated as minima and the symmetrical error bars in Figure 7 are not appropriate. The upper bars should be much longer.

Secondly, the three different measurement techniques (SEVIRI, MODIS, IASI) produce results that disagree by more than the size of the error bars. If two measurements

of the same feature disagree, then either the measurements are wrong or the errors have been underestimated. The reasons for disagreement are not fully investigated, but it is suggested that IASI has higher readings because it works better when the ash is overlaid by meteorological cloud. This suggests that SEVIRI and MODIS have systematically underestimated the mass loading and so again, the upper error bars should be extended. This also means that mass loading estimates based on SEVIRI or MODIS data for other eruptions where clouds were present, e.g. Eyjafjallajökull 2010, were also underestimated.

There are also two extra sources of error that have been highlighted since the references cited made the estimate of 40-50%.

Firstly, the 40-50% figure assumes that the particles are dense spheres. Prata and Prata (2012) cite Wen and Rose (1994) and Gu (2005) [who, in turn cites Gu et al 2003] as the source of this estimate. Wen and Rose only considered the effect of varying the size distribution and particle composition. Gu et al add variations in surface and cloud temperatures. They also include a shape factor, however their study looks at desert sand and not volcanic ash.

The dense spheres assumption is not met by volcanic ash. The authors demonstrate this with CALIOP data, which show high volume-depolarization ratios for clouds rich in volcanic ash. Deviation from this assumption increases the uncertainty. Kylling et al (2013) demonstrated that if the particles are vesicular spheriods, then the mass loadings are underestimated by up to 40%. In fact, the grains are bubbles and plates and shards (see pictures in Stevenson et al., 2013), so the deviation is even greater.

Secondly, satellite retrieval algorithms have been shown to systematically underestimate the grainsize and the mass loading of ash clouds, even where the dense spheres assumption is met. Stevenson et al. (2015) used simulated satellite images to show that algorithms prefer solutions with low concentrations of smaller particles (which exhibit a strong brightness temperature difference effect) to ones with high concentrations

of larger particles (which exhibit a weak BTD effect). Clouds with many large particles do not exhibit a BTD effect, so they are indistinguishable from meteorological clouds and are not included in mass loading estimates. Stevenson et al. (2015) suggest that over 50% of the mass can be missed in this way.

There is evidence of such underestimation in this study. Figures S4 and S5 include effective radius estimates generated from SEVIRI and MODIS data. There are no values higher than 8 microns, which corresponds to a number size distribution with a modal diameter of 5 microns and a 95th percentile of 15 microns. This is significantly lower that measured values from ash deposited in Scotland (median length 19-23 microns, 95th percentile 42-51 microns). On the other hand, the corresponding mass size distribution has 95th percentile of 64 microns. This discrepancy should be discussed.

**4.4 Discussion of the failure to detect ash grains in the upper plume**

Pollen sensor data demonstrates that there was some ash in the upper plume (Stevenson et al., 2013). This is corroborated by reports of ashfall in northern Iceland described in the paper. It is also to be expected, unless the aggregation process was 100% effective.

Some discussion should be included about why it was not detected by the satellites. Reasons may include:

- The ash grains were not spheres, but shards. Although they are 3-5 microns in diameter, they are a tiny fraction of micron thick and may not exhibit a strong differential absorption.

- The low Earth surface temperature of Arctic sea ice and the Greenland ice sheet may limit the potential for brightness temperature differences.

- The ash grains may have been coated by ice.

• The ash concentration may have been too low.

**5   Other modifications**

**5.1   Plume height time series**

I think that the paper would be improved by replacing Figure 3 with a time series of plume height data. Four sources of plume height measurements are mentioned in the paper, radar, cloud shadow, IR brightness temperature for volcanic ash, and IR brightness temperature for SO2 (which appears to be a new technique). It would be very instructive to present a time-series graph of plume height using the various methods. This would allow the evolution of the eruption to be understood and so the height-estimation methods to be compared.

**6   Line-by-line comments (page:line)**

• 2:20 - Magma composition is important factor in separation. Basaltic magmas (e.g. BárÃřarbunga 2014) have low viscosity and fragment into coarse particles that easily separate from gas. Silicic magmas (e.g. Mt St Helens 1980, Chaiten), or basaltic eruptions through water (e.g. Grímsvötn), fragment into smaller particles. Some of these particals are essentially aerosol and only separate from the gas through aggregation.

• 5:20 - Good to emphasise the importance of getting the correct height.

• 7:20 - What is the explanation for the difference in height estimate between radar and that satellite measurements?

- 8:table1 - Add columns for spectral range e.g IR, UV and whether active or passive.

- 9:figure3 - A time series plot would be really nice here.

- 10:5 - Much of this information should be in the figure caption.

- 11:figure5 - (a) add direction of travel arrow to overpass. Which bands are compared in BTD? (d) what is the peak at 5 km? it is as large as the SO2 signal over Greenland. What do the black lines represent?

- 12:5 - The non-spherical particles measured here contradict an important assumption in the ash retrievals and dispersion models. The implications of this should be discussed.

- 14:5 - The definition of fine ash is discipline specific. Ash <63 microns is 'extremely fine' to a volcanologist.

- 14:12 - Define exactly what mass means e.g. sum of all detected, ash-containing pixels?

- 14:figure7 - The error bars appear to be +/-20-25%. This is an underestimate.

- 15:5 - Be explicit about what the different assumptions between IASI and SEVIRI retrievals are, including references.

- 16:9 - The 'Separation of the dispersing volcanic cloud' section is poorly named. It mainly describes cloud top height measurements, so the contents should be moved to that section of the paper.

- 16:15 - Which wavenumbers, which models. Give reference(s).

- 16:17 - Give examples of other causes of positive BTD (with references) and explain why they are 'generally unusual'.

- 16:27 - What are the altitudes of the sounding level peak contributions?

- 18:13 - A pyroclastic flow is typically used to mean a dense current supported by particle-particle collisions moving on the ground. These can travel in all directions depending on topography, including upwind. The more modern and generic term 'pyroclastic density current' includes low density currents supported by a turbulent gas phase. Their travel direction may be influenced by wind. Lenses of different-sized grains in the deposits from G2011, which correspond to dunes or ripples, are evidence that this took place.

- 18:figure10 - The vertical column is shown as the source of a gravity current moving along the ground. What is the source of particles at 4 km altitude that are carried to the UK? Note that images from G2004 in Jude-Eton et al show particles raining downwards from bent over plume then spreading horizontally across the glacier surface in a dilute pyroclastic density current.

- 19:1 - How can a low-level ash layer 'persist' for 24 hours before moving south or east? Surely it would sediment out or be advected away during that time?

- 19:5 - The term 'collapse' here evokes images of total column collapse and dense pyroclastic density currents moving in all directions (e.g. the Pinatubo eruption in 1991). This did not happen at Grímsvötn in 2011.

- 19:10 - The presence of hail is an indication of the removal of water from the plume. The ash itself, particularly the fine grainsize, is the evidence for the rapid removal of ash.

- 19:16 - I prefer the idea of the column sloughing to it collapsing. What is the altitude of this skirt? Does it reach from the ground to 4 km or does it disperse above the ground level.

- 20:5 - The radar measurements should be in the plume height section.

- 20:10 - Be explicit about how the observations rule out ash separation at the source or in the laterally intruding cloud.

- 22:figure13 - Mark a 'Region of wet aggregation' on each of the subplots based on subplot (b).

- 23:11 - Separation of material is not inevitable. Extremely fine ash produced by explosive eruptions is suspended in turbulent air. Separation only occurs if particles aggregate. The composition of the magma and the size distribution of tephra at the vent are crucial. All eruptions are not equal.

- 23:23 - The plume model does not reproduce the observations, it provides evidence for liquid water in the plume that would enhance aggregation.

- 24:figure14 - This chart only shows the presence of ash, and does not mention concentration. It shows the ash-rich plume moving south. It shows the SO2-rich plume, which also contained some ash, moving northwards. This is in agreement with the results presented in this paper.

- 24:1 - Provide a reference for concentrations in Europe. Prata and Prata (2012) present ground-based PM10 data from 6 locations in Norway and Scotland. These cannot be generalised to the atmosphere, nor to a wider area.

- 24:4 - This wording suggests that there was nowhere that the ash was sufficiently concentrated to be a hazard to aircraft. This is not true. The south coast of Iceland during the eruption was clearly hazardous. It is important to specify a location and time. The satellite retrievals presented in figure S5 show total column loadings of around 1.8 g/$m^2$ to the west of Scotland on 23 May. Assuming that the plume was 1 km thick, this corresponds to 1.8 mg/$m^3$. Taking into account the uncertainties in the method that mean these retrievals represent minimum values, then it is probable that there were locations where the concentration exceeded

the 2.0 mg/$m^3$ limit for Zone of Medium Contamination that aircraft require special permission to enter.

- 24:15 - See also recent work by Julia Eychenne on the Mt St Helens pyroclastic density currents as a source of ash plumes.

- 25:2 - I would also argue that the fourth-power law is inappropriate for eruptions through water or ice.

- 25:14 - Is hemispheric spread likely? I thought that circulation patterns moved air to the poles and Grímsvötn is already close to the Arctic.

- 25:15 - Can you put the mass released into context with other eruptions that did effect climate?

- 27:equation1 - What is sgn?

- 27:26 - The sensitivity of fall velocity to the drag coefficient is discussed. The particle density will also vary greatly. Volcanic glass is 2 to 3 times more dense that water, and dry aggregates contain large propotions of void space. This may have a greater impact on fall velocity.

- 28:1 - Why did you use radar plume height data when the rest of the paper disagrees with it?

- 28:16 - It would be excellent to get particle size data for the grains within the hailstones from Arason et al (2013) and compare with the predictions. Is it possible to contact them?

- S1:6 - Why only model up to 10 km when the plume went higher?

- S2:2 - What is the geometric standard deviation of the distribution?

- S2:5 - Give more information about the bi-Gaussian vertical distribution. What were the levels, what proportion of mass was assigned to each, why were those values chosen?

- S7:figureS4 - What is the geometric standard deviation assumed?

- S8:figureS5 - What is the geometric standard deviation assumed?

**7 References**

- Gu Y (2005) Advantageous GOES IR results for ash mapping at high latitudes: Cleveland eruptions 2001. Geophysical Research Letters. doi: 10.1029/2004GL021651

- Gu Y, Rose WI, Bluth GJS (2003) Retrieval of mass and sizes of particles in sandstorms using two MODIS IR bands: A case study of April 7, 2001 sandstorm in China: RETRIEVAL OF MASS AND SIZES OF PARTICLES IN SANDSTORMS. Geophysical Research Letters. doi: 10.1029/2003GL017405

- Jude-Eton T, Thordarson T, Gudmundsson M, Oddsson B (2012) Dynamics, stratigraphy and proximal dispersal of supraglacial tephra during the ice-confined 2004 eruption at Grímsvötn Volcano, Iceland. Bulletin of Volcanology 1–26. doi: 10.1007/s00445-012-0583-3

- Kylling A, Kahnert M, Lindqvist H, Nousiainen T (2013) Volcanic ash infrared signature: realistic ash particle shapes compared to spherical ash particles. Atmos Meas Tech Discuss 6:8937–8958. doi: 10.5194/amtd-6-8937-2013

- Prata AJ, Prata AT (2012) Eyjafjallajökull volcanic ash concentrations determined using Spin Enhanced Visible and Infrared Imager measurements. J Geophys Res 117:24 PP. doi: 2012 10.1029/2011JD016800

Stevenson JA, Loughlin SC, Font A, et al (2013) UK monitoring and deposition of tephra from the May 2011 eruption of Grímsvötn, Iceland. Journal of Applied Volcanology 2:3. doi: 10.1186/2191-5040-2-3

Stevenson JA, Millington SC, Beckett FM, et al (2015) Big grains go far: understanding the discrepancy between tephrochronology and satellite infrared measurements of volcanic ash. Atmos Meas Tech 8:2069–2091. doi: 10.5194/amt-8-2069-2015

Wen S, Rose WI (1994) Retrieval of sizes and total masses of particles in volcanic clouds using AVHRR bands 4 and 5. J Geophys Res 99:PP. 5421-5431. doi: 10.1029/93JD03340
* * *

---

## Referee Comment (RC2) · A. Folch (Referee) · 13 Mar 2017

Atmospheric processes affecting the separation of volcanic ash and SO2 in volcanic eruptions: Inferences from the May 2011 Grímsvötn eruption" by Fred Prata et al.

This study uses multiple Infra-red, Visible and UV satellite data to illustrate the separation of volcanic ash and SO2 occurred in the eruption column of the May 2011 Grímsvötn eruption (Iceland). The observational evidences are complemented with atmospheric dispersal and plume model simulations to elucidate the mechanism leading to separation, here attributed to the formation in the plume of large hydrometeors (ice-coated ash aggregates) eventually leading to partial column collapse. This resulted on ash and SO2 emitted at different atmospheric levels which, combined with strong wind

shear conditions, explains the contrasting dispersal patterns. A part from using and comparing an unusually large variety of instruments (AVHRR, MODIS, AIRIS, OMI, SEVIRI and Caliop), the paper supposes also a contribution to: i) warn on the use of SO2 as a proxy for volcanic ash and, ii) claim on the need to implement differentiated source terms for ash and SO2 in atmospheric dispersal models, an option currently not contemplated by any model (to my knowledge). This is a good paper which needs minor revision; I have only few comments detailed below.

Ash aggregation and hail formation provide a plausible mechanism for separation, supported by fallout deposit observations (Fig. 11) and integral plume model insights (existence of a liquid water window). However, the column collapse scenario (Fig. 10) is more speculative (pg. 18, lines 12-14). It is also unclear how is fine ash (not large millimetric size aggregates) was released at low levels. . .

Pg. 3, line 1. "The separation led to a poor forecast of the ash hazard to aviation". It would we worth to clarify why this may happen. In general, if: i) observed column height and ash are assumed collocated or if, ii) SO2 retrievals are used to initialize a dispersal model.

Pg.5, line 9. atmosphere–so-called atmosphere so-called

Pg. 6. Line 4. Reference Degruyter and Bonadonna (2012; 10.1029/2012GL052566) could be added.

Pg. 7, line 1. "The initial amount of erupted fine ash required to generate an ash cloud consistent with the satellite observations cannot be modelled using the fourth power law, as this produces almost 100 times too much fine ash". The H4 scaling applies to the whole granulometric spectrum, not just to the fine ash fraction (typically a few %). The mass scaling may be needed because the model run with a very fine skewed granulometry.

Pg. 7, line 16. ". . .such as low to medium size volcanic eruptions". This limitation of

CALIPSO is more related to the duration of the event than to the eruption size no?

Pg. 11, Fig. 5 caption. "ornage" orange

Pg. 12, line 15. figure Figure

Pg. 16, line 6. "which is an order of magnitude greater from an eruption that was an order of magnitude smaller in total erupted mass than the Grímsvötn eruption". Section 2 reports 0.7 and 0.27 km3, for Grímsvötn and Eya respectively, which is a factor of about 2 only...

Pg. 24, line 11. Reference Folch et al. (2016; doi:10.5194/gmd-9-431-2016) could be added.

Pg. 24, Fig. 14 caption. A third contour (blue) is also visible

Pg. 28, line 5 and line 22. Table number.

Figure 15. Time series unclear. Plume top is a model prediction or imposed from observations? Also, is the plume model wind coupled?

---

## Author Comment (AC1) · 15 May 2017

**Response to reviewer 1**

We thank John Stevenson for a thorough and thoughtful review of our paper. He raises many interesting and helpful issues and here we address these.

**General**

Many of his comments are directed towards two main areas:

[Figure]

1. Column collapse and aggregation.

2. Errors and their sources in satellite estimates of mass loadings.

In both of these areas he refers to his own body of published work and asks us to consider our results in the light of his own results. Our rebuttal begins with his Section 2.

**Section 2: Contribution**

Stevenson argues that some statements are not well supported and should be dropped. Our item-by-item response follows.

1. We agree with this statement and have added a statement along these lines in the conclusions.

2. "Separation of material". It is difficult to imagine a case in the atmosphere where wind shear would not lead to horizontal separation. Perhaps, Stevenson interpreted our findings to mean vertical separation? We have modified our statements to make it clear we are referring to horizontal separation of gas and particles.

3. We have modified our description of column collapse and acknowledge that the processes we describe may not fit well with the standard description of a column collapse. We did preface our description with use of the word "partial", but have now also explicitly described the observations more in line with Stevenson's suggestions.

4. Stevenson seems to suggest the ash cloud was a significant threat to aviation. This does not agree with the observations. There were no reported aviation incidents and the ash was not of high enough concentration when it crossed into aviation flight routes. Note that flights across the North Atlantic/North Sea typically use altitudes >20,000 ft (except at take-off and landing). The dispersing ash cloud was confined to altitudes below 20,000ft.

5. We have added the words "under the assumption of dry standard atmosphere conditions" when referring to the Mastin relationship.

**Section 3: Suitability for publication**

We have made the necessary revisions as described in this rebuttal.

**Section 4: Areas for revision**

*4.1 Missing reference.* We have included a reference to Stevenson et al. (2013). The first two comments in this Section need no comment because they are statements and do not seem to contradict our paper.

*Ash deposited in Scotland.* The third comment concerning the deposition of ash of 19-23 $\mu$m median grain size with maxima of 80 $\mu$m is interesting but largely irrelevant to our paper. Without going into a detailed analysis of Stevenson et al.'s results we simply state from his own paper: "The sizes of measured grains had modes of 25–30 $\mu$m although grains <10 $\mu$m were not counted, and those close to this minimum size were more likely to be missed." This kind of sampling bias is sufficient to reconcile any differences between the two independent methods of estimating

particle sizes. Stevenson et al.'s methodology counts no particles with sizes <10 $\mu$m and the satellite retrieval scheme detects only particles with sizes <10 $\mu$m. However, we stress that the satellite is measuring within the atmospheric plume and needs to make no assumptions about how the particles reached their location. While the infrared sensors discriminate ash on the basis of the BTD signal, and the sensitivity of that technique relies on small particles (effective radii <16 $\mu$m or so), the detection of particles relies on detecting a change in optical depth. This difference between detection and discrimination is critical. Large particles in sufficiently high concentration will cause a detectable change in optical depth and hence will be detected by the infrared satellite sensors. It is nearly always the case that these particles reside in the most optically dense part of the plume and that part is generally closest to the source. A dispersing plume loses mass by deposition (mainly) and the large particles are removed. One can imagine scenarios where large particles co-exist with high concentrations of small particles and are somehow hidden from discrimination (but not detection). We admit such cases may occur but so far there has been no evidence presented of these scenarios. The occurrence of large grains deposited in northern Scotland is consistent with the satellite retrievals that suggest only the small particles remain in the plume. The route taken by these large particles may not be the same as the route followed by the small particles. Indeed it is possible that the large particles have arrived via a different, more tortuous route, re-cycled by the atmosphere and advected by winds at different heights. In any case, we do not agree with Stevenson et al.'s assertions that larger particles exist in plumes and go undetected by a variety of different sensors (satellite IR, active lidar, in situ aircraft particle counters), which we feel are unproven. In contrast, there is evidence from independent research that is in strong support of our analysis. Moxnes et al. (2015), for example, show aircraft measurements of the particle size distribution near the   plume on 22 May (see their Figure 15). The particle diameter measured (and modelled) ranges from 1–20 $\mu$m with a peak in the distribution of $\sim$5 $\mu$m. This is in broad agreement with our satellite retrievals (r effective of 5–7 $\mu$m). This leads us to conclude that the methods adopted

by Stevenson et al. are largely incompatible with measurements of particles in plumes. There is a clear need for a scientific study to fully understand the causes of these disparate results and inform the interpretation of measurements of grounded particles in conjunction with satellite measurements of suspended ash particles.

*Pollen traps and trajectory analysis.* It is hardly surprising that small (tiny?) particles with sizes of 3–5 $\mu$m were detected in raindrops, after all these are the size range for particles that the satellite IR sensors detect (and detected in this case). The trajectory analysis is however, suspect. Figures 1a and 1b of Stevenson *et al.*'s paper seem to show that the southern trajectories that bring air over Scotland begin at low altitudes. The air that passes over Greenland stays there or moves north, west or east (Stevenson *et al.* comment that trajectory analysis is consistent with smaller particles arriving in the UK via Greenland on 27 May, but no data were provided). We feel this needs no further comment as our rebuttal is not concerned with the interpretations of Stevenson *et al.*'s body of research. On the point about why the IR retrievals did not detect ash over Greenland – we simply note that they did detect some ash and this has been reported elsewhere (Prata and Rose, 2015; see Figure 52.11).

*4.2 Discussion of plume "collapse".* We agree that our interpretation is speculative and have tidied up some text (including terminology) as recommended by Stevenson. We have also added a citation to Jude-Eton *et al.* (2012).

*4.3 Uncertainty estimates for ash retrievals.* Stevenson goes into some detail to suggest that satellite retrievals are much less certain than we have stated. We think in general this may be true and there are many assumptions relied upon, which might not be appropriate in all cases and could be improved with further study, and these contribute to the uncertainty in the retrieval. However, this paper deals with one case, the 2011 eruption, where we are more certain of the error estimates used. Moreover, rather than requiring a higher error, the validation data for suggests the satellite

retrievals have a lower error than the conservative estimates we have adopted. Prata and Prata (2012) report air quality validation data for the SEVIRI satellite retrievals (see Figure 18 of that paper). Four of the five validation points show the satellite retrieval is higher than or close to the PM10 measurement. Only one point is lower. It is also the case that validation data for Eyjafjallajökull using the same retrieval scheme gives results within $\pm 0.2$ mg m$^{-3}$ of independent data. For a retrieval with concentrations of 0.2 mg m$^{-3}$, the accepted lower limit of detectability for current IR broadband imaging sensors (hyperspectral sensors may do better), the error is $\pm 100\%$. But for concentrations that matter to aviators ($>2$ mg m$^{-3}$) the error is $\pm 10\%$. Stevenson *et al.* (2015) relies on simulated imagery, which is subject to greater error than actual observations. Here he repeats the common misconception of the way the IR sensors are used to detected ash clouds. Clouds (ash or otherwise) with high concentrations of large particles are always detected by the IR sensors. The retrievals cannot be confidently performed when there is no BTD signal, but satellite retrieval practitioners and experts do not claim to retrieve mass loadings in these optically thick parts of the cloud. The mass loadings are provided on a pixel-by-pixel basis and vary across the plume. There is no underestimation (or overestimation) within the error bounds provided for these retrievals, and for the optically thick parts or where meteorological cloud interferes, no retrieval is made. So as with any measurement system, when the conditions under which the retrieval methodology is met, an error estimate can be provided. In optically thick clouds there is no means to provide an error estimate. This comment also needs to be stressed with regard to other assumptions, for example dense spheres and the presence of meteorological clouds. In the case of the dense sphere assumption, the "sphere" part is used for Mie calculations and the "dense" part is a scaling factor. If the ash density assumed is incorrect by 50% then so is the mass loading. The retrieval methodology used is state-of-the-science and further study is required to improve the assumptions. Information on odd shaped particles, shards, asperities, bubbles, density, compositional uncertainty are not readily available and we use what is currently accepted. One could assume all ash particles are plates

and shards and large (grain size $>32$ $\mu$m), but the retrievals would not agree with the validation data. We note also that dispersion models rely on similar assumptions concerning the density, shape and size distribution of particles in order to get accurate transport.

With regard to the last comment by Stevenson in Section 4.4, we see no discrepancy here. As explained earlier, the ash deposit measurements sample a different portion of the size distribution to that sampled by the IR sensor. In fact we would argue that the ash fallout is irrelevant to our study as necessarily this is ash that is not in the plume (it has fallen out – recall if it were still in the plume then the IR optical depth would be affected).

**4.4. Discussion of failure to detect ash grains in the upper plume.**

This is a curious comment because the IR sensors do detect ash in the northern part of the plume. Stevenson *et al.* (2013) do not demonstrate that pollen sensor data show that ash was present in the upper part of the plume. At best it is speculative based on a trajectory analysis (that was not shown). Their own analysis shows it was the southern part of the plume that crossed Scotland, and this is also shown in our paper. With regard to general mechanisms for lack of detection of ash by IR sensors, there are many, of which Stevenson suggests some. We do not agree with the suggestion that "low Earth surface temperature" will limit the potential for BTDs. It is not the surface temperature that matters; it is the difference between the background temperature (in the case of satellites it is usually the Earth's surface, but could be a cloud deck), and the temperature of the ash cloud or plume. If the surface is colder than the ash cloud, a BTD is still detected but has the opposite sign (see Becket *et al.*, 2017 who make use of this for remobilized ash). When the thermal contrast between the ash and the underlying background are the same, the BTD vanishes and no ash discrimination or

quantification is possible. In any case this northern branch of ash was detected and the signal soon dissipates – probably because the ash was of low concentration and soon falls below the detection limit of the sensor.

**5. Plume height time series.**

Time series of plume height are already published by Petersen *et al.* (2012) Figure 5 and Moxnes *et al.* (2015). It is difficult to justify repeating that data in our paper, when it is already clearly presented, accessible and referenced in our paper.

**Line-by-line comments.**

**2:20** We have added a comment on this.
**5:20** Thanks.
**7:20** Both methods have shortcomings. The radar has a vertical resolution of between 2–5 km at a range of 75 km (see Figure 2 of Petersen *et al.*, 2012) and the satellite method depends on pixel resolution and of course solar and satellite viewing geometry, and can be $\sim\pm1$ km. The differences between the two methods are within their respective error uncertainties.
**8: table 1** Done.
**9: figure 3** We agree that a new plot of the height time series would be nice but as this is already published (see Figure 5 of Petersen *et al.*, 2012 and Figure of 3 Moxnes *et al.*, 2015) we find it difficult to justify.
**10:5** We have moved some to the Figure caption as suggested.
**11: figure 5 (a)** We can't see the relevance of the direction of travel of the satellite, but have added that information in the caption. The paper by Prata *et al.* (2016) describes the AIRS BTD technique (several bands are used). (d) the peak at $\sim$5 km is also ash

(see Figure 5(c)). This is ash over the sea ∼64 °N – it is not over Greenland and is not sulphate. The black lines show the height range over which the parameters were calculated. We have added an explanation in the caption.

**12:5** Comment added in text.

**14:5** Agreed. We have defined our meaning of "fine".

**14:12** Agreed. This has been added to the text.

**14: figure 7** This is not an underestimate.

**15:5** References added.

**16:9** We disagree and have left this unchanged.

**16:15** References added.

**16:17** The causes of positive differences are mostly associated with water vapour and thermal contrast effects. There is no other literature that we are aware of discussing positive differences in this product. The point of the sentence is to emphasize that the strongly positive anomaly over   is unusual.

**16:27** This is very difficult to state without including a large amount of explanatory text. Since there are more than 2000 sounding channels, there are over 2000 level peak contributions. We have added a general reference on this topic which concerns remote sounding and information content.

**18:13** Yes we were sloppy in our use of terminology and have changed to PDC and added a reference to a discussion of the different uses of these terms.

**18: figure 10** It is difficult to unequivocally assign a source to the ash that is transported in the atmosphere at low altitudes. We hypothesise that the source of the ∼3 km ash that eventually found its way to Scandanavia is a combination of ash from the partially collapsing column and some ash fallout from above. We note also that the image of the eruption column in Figure 1 shows ash separating from the main column above the ground but at low altitudes. There is no strong evidence of PDCs in the deposited material, suggesting much of the ash separating from the column did not feed PDCs on the ground. Unsteadiness of the column and the source could also contribute ash at low levels.
**19:1** Of course we do not mean to say that the ash is static during this period, and we expect that atmospheric transport processes (e.g. buoyant transport, advection by the low level winds, particle settling) are acting on this ash cloud. We note that the winds were not strong and so the ash moved slowly. The cloud would also have been fed by new ash from the on-going minor eruptions. It can only be advected away if there are winds to do the advection.

**19:5** We agree that the term 'collapse' is commonly related to total column collapses, but note that we preface this with partial collapse or even collapses. We have added a sentence to clarify this picture.

**19:10** We have modified the sentence to read "The photograph shows a short vertical section dug into the deposit with evidence of hail that is collocated with ash, and suggesting removal of water."

**19:16** Yes this is a good alternate term and we have added it to the text.

It is difficult to estimate the altitude of the skirt as the brightness temperatures can't be used and there is no shadow. It is unlikely to be higher than ∼4 km, although there could be a lower concentration of ash above (undetected and likely undetectable). Again there is not sufficient data to state the lower height limit of the "skirt" and it may extend to the ground.

**20:5** Yes we have moved this part.

**20:10 Figure 1** is very direct evidence that separation is occurring from the convective column.

**22:figure 13** We have added a shaded region to each panel indicating the presence of liquid water. It is probable that aggregation can also occur above this region, where all condensed water is expected to be ice, but we expect the rate of aggregation to be higher when liquid water is available.

**23:11** We are discussing separation of gas and particles. We have rephrased this as: "The vertical separation of gases and particles in volcanic eruption columns occurs frequently and if it occurs in the presence of wind shear it is inevitable that this results in a lateral separation of gases and particles distally. Wind shear is ubiquitous and significant when eruption columns extend to the tropopause and consequently it should
be expected that some separation will occur."

**23:23 Agreed** Sentence changed to: "The plume model that we use here to analyze
the transport of particles in the eruption column highlights the importance of multiphase
processes, particularly the role of water in vigorous eruption columns."

**24:figure14** No. The VAG shows clearly that ash was forecast over the Norwegian
sea up to FL200 (20000 ft or 6-7 km). There is no evidence of ash at that location in
the observations. The VAG does not agree with the observations. This is not a trivial
academic point. Closure of airspace there prevented helicopters from reaching oil-rigs
and had an unnecessary economic impact. Stevenson's comment highlights an impor-
tant concern, that scientists looking at the same graphic can make completely opposite
inferences. This demonstrates the (intentionally) low information content of the VAG.

**24:1** Three references provided. But actually they can be generalized because the
ash was detected in Norway, Scotland, Sweden, Holland and northern Germany. So
everywhere else there was less concentrated ash or none.

**24:4** The wording implies northern Europe and needs no modification. Stevenson's
argument about ash concentrations contradicts the evidence. The PM10 stations show
concentrations $<500$ $\mu$g m$^{-3}$. Why assume a thickness of 1 km? Why not 3 km? Then
the values are more like 300 $\mu$g m$^{-3}$, which agrees with the observations. Tesche et
al. (2015) also report lidar measurements in the range 150–340 $\mu$g m$^{-3}$. Moxnes *et al.*
(2015) report values $< 100$ $\mu$g m$^{-3}$ based on aircraft data and modeling. These data,
our data, previous measurements from Eyjafjalljökull (lidar, airborne and ground-based
air quality) all provide adequate support for the assumptions we use in satellite-based
IR retrievals. Our statements match closely to what was observed. Our error estimates
for the eruption are robust and should not be extended to all ash retrievals for any
other eruption.

**24:15** Thanks.

**25:2** Agreed.

**25:14** It is less likely to cross hemispheres or even reach low latitudes from a high

latitude eruption, but zonal dispersion is likely. Hemispheric spread (in this context meridional and zonal in one hemisphere) is more likely, the longer the aerosol remains in the atmosphere. This was the point of the sentence.

**25:15** Sentence added.

**27:equation 1** sgn signifies the mathematical signum function. It is standard notation, -1, if $x < 0$.

**27:26** Agreed. We note that the variation in density of solids (from $\sim$700 kg m$^{-3}$ for vesicular pumice to $\sim$3200 kg m$^{-3}$ for glass shards) does not greatly alter the critical fall-out velocity calculated using our reference density (1200 kg/m3) with changes in the value by a factor of 0.76 to 1.6.

**28:1** The rest of the paper does not disagree with the radar heights.

**28:16** Arason *et al.* (2013) have presented this data at a scientific conference and have made this presentation available through their personal website. The ash in ash-infused hail ranges in size from 2 microns to 10 mm with a peak for 0.2–1 mm. This might form a useful later study.

**S1:6** Because we are only modeling the "ashy" part. There was very little, if any, ash from the upper part of the plume that was transported southwards.

**S2:2** It is a unimodal particle size distribution.

**S2:5** Standard parameters were used as specified in the FALL3D documentation, but as mentioned many runs were used.

**S7:figure S4** There isn't just one geometric standard deviation assumed. The scheme generates a large number of brightness temperature simulations based on Mie calculations that assume different effective particle radii and geometric standard deviations for the lognormal distribution. The procedure finds the best fit between the observed and simulated brightness temperatures.

**S8:figure S5** As above.

**References**

Becket, F., Kylling, A. (2017) Quantifying the mass loading of particles in an ash cloud remobilized from tephra deposits on Iceland, Atmos. Chem. Phys., 17, 4401–4418.

Prata, A. J. and W. I. Rose (2015) Volcanic ash hazards to aviation, Encyclopedia of Volcanoes, 2nd Edition, pg. 911–934.

Tesche,M., P. Glantz, C. Johansson, M. Norman, A. Hiebsch, A. Ansmann, D. Althausen, R. Engelmann, and P. Seifert (2012) Volcanic ash over Scandinavia originating from the   eruptions in May 2011, Journal of Geophysical Research, 117, D09201, doi:10.1029/2011JD017090.
* * *

---

## Author Comment (AC2) · 15 May 2017

**Response to reviewer 2**

We thank Arnau Folch for his comments.

**General**

He is correct in saying that our hypothesis of partial column collapse is speculative, but we feel it is a plausible explanation that the data supports. In light of the comment

we have emphasized the speculative nature and included some alternate possibilities, including the idea that ash aggregation and fall-out could have contributed to the "sloughing" of the ash curtain without a column collapse.

In terms of how the fine ash was released at low levels, we think that a more likely mechanism is that the partial collapse of the column or "sloughing" due to aggregates falling through the column and reducing vertical thrust, would have been sufficient to prevent fine particles from reaching greater heights. It seems unlikely, according to the observations, that a new source of fine particles was generated by a secondary smaller eruption (although such eruptions may have contributed to ash generation later)– which could also explain fine particles at low levels. The presence of fine particles is confirmed in the observations so there is no argument that they were not present. We have added a sentence to discuss the likely origin of the fine particles at low levels.

**Minor comments**

**Pg3. Line 1**. The point here is that without considering dynamical processes happening inside the column, such as aggregation and fallout, column collapse, or "sloughing", current models may misrepresent the total mass of fine particles in the plume. Current satellite observations can distinguish between a volcanic plume consisting mostly of $SO_2$ and one consisting mostly of ash, but most dispersion models are initialized with either one or the other of these constituents. This is critical at the start of an eruption, where it is more likely that both constituents are present and subsequent separation is likely. In general, most forecasts are good. But without considering separation, which implies that both sources should be modeled, there is potential to forecast the movement of a volcanic ash cloud, when in fact it is largely an $SO_2$ cloud. This seems to be what happened during Grímsvötn. In fact two problems arose: first the cloud moving north and the spreading eastwards and westwards was predominantly $SO_2$ and very

high ($>$ 10 km); secondly, the ash cloud moving south and then eastwards towards Scandanavia, was low and much less "massive" than believed.

**Pg5. Line 9.** Corrected.

**Pg6. Line 4.** Reference added.

**Pg7. Line 1.** Our sentence was misleading and we have corrected it according to the comment.

**Pg7. Line 16.** Yes, I guess that is strictly true, except larger eruptions emit more particles and gases and these spread more and for longer giving more possibilities for an instrument in polar orbit with a very narrow swath to observe them. A case can be made for having more lidars in space.

**Pg11. Figure 5.** Correction made.

**Pg12. Line 15.** Correction made.

**Pg16. Line 6.** We were referring to the mass of fine particles and have corrected the last part of the sentence to "...which is an order of magnitude smaller in erupted mass of fine particles than the eruption."

**Pg24. Line 11.** Reference included: this was an oversight.

**Pg24. Fig. 14.** Caption changed to "from the surface to three flight levels: FL200 (20,000 ft in red), FL350 (35,000 ft, in green dashes), and FL550 (55,000 ft blue dots)."

**Pg28.** Line 5 and Line 22. Corrected.

**Figure 15.** The plume top altitude from the radar time series is imposed to allow us to compute the source condition required to produce a plume rising to the observed altitude. The plume model includes the effect of wind and other atmospheric conditions (temperature, pressure, humidity).

---

## Author Response (AR2)

**Rebuttal**

Dr Fred Prata, 6 July 2017.

The revisions required do not seem to be of great significance to my paper. There seems to be a desire to have more statements on increased uncertainty in retrievals. My paper is not claiming any particular improvement on accuracy of retrievals or retrieval methods, in general. The results presented are for Grímsvötn and were validated. But the paper does not depend in any way on exactly how accurate they are.

**Responses**

*"I would like to see more detail on the uncertainties due to cloudiness and lack of thermal contrast. In the Response to Reviewers, the authors suggest that I am under the 'common misconception' that ash clouds with high concentrations are not detected. They are detected as some kind of cloud due to their increased optical depth and lower brightness temperature, but cannot be identified as volcanic ash and a retrieval cannot be made. What is not clear from the current manuscript, nor from other similar publications e.g. Prata and Prata (2012), is how such pixels are identified and incorporated into mass loading estimates. Is it a manual process? Are they excluded from calculations? In which case, mass loadings must be considered minima."*

**Response:** There are plenty of papers and technical reports that go into detail about ash detection – which seems to be the reviewer's concern here. The interested reader can find a series of technical reports on ash detection, validation and errors here: http://vast.nilu.no/project/deliverables/. It is not the purpose of this paper to examine ash detection methods in any detail. This would require a very major revision of the m/s and a complete change of emphasis. Suffice to say the Grímsvötn retrievals were validated and found to fall below the commonly accepted errors of 40-50%. Kylling states 45-50% discrepancy for shape but really we don't know how this compares to observations because it is a theoretical study. This could all be bias and could cancel with other biases – it is simply not known. What he means is that if you use some irregular shapes then you can get 45-50% differences in retrieved mass loadings – but what shapes should one use? What composition? (Composition and shape are correlated). Has any of it been validated?

In my last response I was quite clear that optically thick pixels are not used in the retrieval and in fact no retrieval is possible. The mass loading is determined on a pixel-by-pixel basis. The retrieval for each pixel could be high, low or just right. Why should they be considered minima? For optically thick pixels no retrieval is made – no statement can be made about whether it is high, low or just right, because no retrieval is made. Likewise, for optically thin pixels, when a retrieval is made it could be too high – in fact that is more likely because often the optical thickness may lie within the noise but the retrieval is made based on other information – for example the spatial context. It cannot be stated that the mass loadings should be considered minima. Perhaps, the total mass retrieved is biased low. There are other complexities. To demonstrate, consider this: for a MODIS pixel ($1 \times 1$ km$^2$) suppose I retrieve a mass loading of 1 g m$^{-2}$. Now I have assumed (implicitly) that the pixel is completely covered by ash – but that's an assumption. It could be that the ash is actually confined to an area of 100 m x 100 m. The total mass then would be 10 kg. But for the MODIS pixel observation, the 10 kg is spread over 1 km$^2$. So the "real" mass loading is 10 kg km$^{-2}$ or $1 \times 10^{-2}$ g m$^{-2}$. In other words I have overestimated the mass loading by a fact of 100. My retrieval is hardly a

minimum.  You could ask if the total mass were actually 10 kg then how could I retrieve 1 g m$^{-2}$?  This is complicated because it depends on what else is in the pixel and exactly where the ash is.  The instrumental response is not uniform across the pixel and indeed since we use two different channels, the fields-of-view (fov) need to be co-aligned (they never are).  If the edge of one fov observes cloud, while the same edge of the second fov does not, then the brightness temperature difference between the fovs could be negative without any ash at all in the pixel.  Of course, the problem here is one of heterogeneity and spatial resolution – I would need to address this as well as all the other complexities, such as sub-pixel cloud, "mixels", misalignment of instrumental field-of-views, calibration non-linearities (these affect "cold" pixels), slant-angle effects, overlapping pixels, the modulation transfer function ... if I were to properly address uncertainties in satellite retrievals.  But see Fig. 14 of Clarisse and Prata (2016) (referenced in this paper) where fovs from three different IR sensors are collocated.  It is a complex problem.

*"Just because someone wins the lottery doesn't mean that the odds weren't millions to one!"*

**Response:** I really don't know the meaning of the comment.  The data are what they are – no luck was involved.  The validation shows that the Grímsvötn retrievals were better than to be expected.  It is a reasonable admission that errors, in general, could be larger.  This is a paper about Grímsvötn ash dispersion.

*"The brightness temperature difference method assumes that particles are dense spheres, which only exhibit the BTD effect when the size distribution is dominated by particles <10 μm diameter. Thus, any pixel displaying a BTD signal will be interpreted as being dominated by particles <10 μm diameter. If nonspherical and bubbly particles cause a BTD signal at larger grainsizes (as demonstrated by Kylling et al., 2014), and ash grains are not dense spheres (as demonstrated by hundreds of tephrochronology studies), then the grainsize will be underestimated."*

**Response:** Actually no such assumption is made.  The reverse observation due to ash happens – it is observed in the data.  The model makes various assumptions but that is a different matter.  The paper is not claiming any significance about grain size retrieval.  Kylling says the range is increased from 5 μm to 10 μm (I think it is larger than 5 μm for dense spheres).  Kylling does not address "a cloud of particles" where the radiation from individual scatterers interacts.  If you look at his Figure 5 you will see that he has brightness temperature differences as low as -20 K.  These are never observed.  It does not match reality.  Further, his particle shapes are also "idealized".  Real particles have asperities, aggregates and are also compositionally complex.  The radiative transfer for these particles may be quite different to Kylling's particles and may even suggest that treating them as dense spheres works just as well if trying to estimate mass loadings. The only way to tell is to validate the theory.  Prata and Prata (2012) did that.  The retrievals for Grímsvötn were validated – the theory could still be wrong (probably is wrong) but my paper is not trying to suggest this nor do the results depend on this.

The "hundreds of tephrochronology studies" are irrelevant to very fine ash dispersing in the atmosphere.  (They are in fact orthogonal studies as they are studies of **exactly** what is not in the atmosphere).

*"The only logical way to argue that this doesn't apply is to present evidence that Kylling is wrong, or that volcanic ash grains ARE dense spheres when airborne."*

**Response:** Well it is absolutely clear that you cannot show that Kylling et al. is right. Indeed in the Karl Popper sense I have already shown that Kylling et al. is not right – I validated the dense sphere assumption against independent data and found it to be reasonable. Their study is theoretical and they provide absolutely no data to validate their findings. They are therefore just one of several possibilities that can only be shown to be wrong by experiment. (It can never be proved to be right). Suppose we use odd shaped particles with vesicles. The retrievals are going to produce 45-50% larger mass loadings and disagree with the ground-based data, the lidar data, the aircraft measurements, and all other satellite retrievals. So some other assumption must be wrong.

Kylling et al. also state that: "*It is noted that ash particle shape is not usually known for an on-going volcanic eruption. Thus, for operational monitoring of ongoing volcanic eruptions it is preferable to assume spherical ash particles and rather increase the uncertainty in the mass estimate.*" "… not usually"???? They mean "never". I have never seen a single study, published or otherwise, that reports particle shapes in dispersing ash clouds. Furthermore, what is the distribution of shapes? This must be important too.

*"Neither the response to reviewers nor the updated manuscript addresses the additional source of uncertainty described in Stevenson et al. (2015), namely that the mathematics behind retrieval algorithms biases them towards solutions involving smaller grainsizes. For a given observation, the algorithms prefer solutions with low concentrations of optically active (small diameter) particles."*

**Response:** The algorithms have been validated. I'm not sure why the reviewer says mathematics? He may mean physics? I have added a paragraph stating what I think the Stevenson et al. (2015) paper is reporting.

*"The method of Prata and Prata (2012) is not a 3-parameter retrieval like that of Francis et al. (2012), but instead uses a lookup table for a specific cloud-top temperature. In this scenario, there is only one possible combination of effective radius and optical depth that matches a given observation. The choice of cloud top temperature can therefore bias retrievals towards higher or lower grainsizes. I would like to see discussion of how the cloud top temperatures were chosen and the contribution of varying this to the uncertainty in the retrievals."*

**Response:** The Prata and Prata (2012) paper underwent peer review and was published after revision. If the reviewer wishes to critique that paper then he should submit a comment to JGR. I'm not prepared to enter into a discussion of that paper and it is unusual to expect this.

I'm not sure I understand the meaning of "…*a 3-parameter retrieval like that of Francis et al. (2012)*"? Does he imply this is somehow superior or correct? The Francis method uses optimal estimation. It is necessary that the parameters being retrieved are uncorrelated. In the case of Francis et al. they are not. They retrieve plume altitude, effective particle radius and mass loading. Two of these are correlated. Plume altitude is extremely difficult to estimate and I would argue is a large source of introduced error into the retrieval scheme. A key point: if you have two measurements then you can only retrieve two parameters, unless you add constraints (assumptions).

I am not at all sure why I am being asked to comment on other papers and other

methods.  This is largely irrelevant to the current paper.

*"There are some reasons why we cannot be sure that aggregation is the sole driver of a partial collapse." I'm not clear what argument you are making here. Are you saying that there are many course particles and so the plume would have collapsed anyway? Can you rephrase?"*

**Response:** Yes this has been re-phrased to:

 "Because there is not that much very fine ash in the column to begin with to generate a sector wide plume collapse we cannot be sure that aggregation is the sole driver. The particles in the 100 μm size fraction contain less than 10% of the mass erupted at any one time, so that even if all of this ash forms aggregates, the mass fraction is still small compared to the total mass."

Minor comments.

**References**
Clarisse and F (2016): who is second author?  Changed to Clarisse, L. and Prata, F. (2016).
**Line by line comments**
3:10 - Did you mean that 16 microns is the largest size?  Yes.  "smallest" changed to "largest".
3:24 - Spelling: glacier  Corrected.
6:9 - Standard atmosphere is important, but the ice/water at the vent at Grímsvötn were probably a bigger factor and are not accounted for in Mastin.  Agreed.  Changed to "…, without the inclusion of ice/water at the vent, which is likely a large factor in the case of the Grímsvötn eruption.
7:30 - Spelling: grain sizes  Changed "size" to "sizes".
17:5 - Do you mean fine ash or very fine ash? The two terms seem to be used interchangeably in consecutive sentences. Clarify by repeating definition.  OK.  I have defined my use of the term very fine ash and replaced all instances of "fine ash" with "very fine ash".
18:6 - Kylling (2014) found errors of 40%, which is greater than the 10-30% cited here.  Kylling found this to be 40% but as this differs from someone else's theoretical study, one is driven to the conclusion that the models have some inherent discrepancy.  I have therefore decided to rephrase this entire sentence to read: "Estimating precision in retrievals is difficult because of the uncertainties in the input parameters, such as the complex index of refraction, the size distribution and the shapes of the particles, although shape is generally found to result in the smallest discrepancy of the input parameters with theoretical simulations showing differences in the range of 10-40% (Yang, 2007, Kylling et al, 2014)."
20:29 - Spelling: Grímsvötneruption Corrected.
22:7 - Adding meltware removes energy from the plume  Changed to "…in the form of hot water vapour (steam) and also contributing …"

**Co-Editor comments**

Errors.  (See my comment above).  I have added this paragraph in Section 4.5:
"Retrieval methods are being continually improved and there is an international effort (http://cimss.ssec.wisc.edu/meetings/vol_ash15/) to inter-compare retrieval schemes

and help reduce uncertainty.  At the current time no firm conclusions have been made about retrieval accuracy as no robust validation has been made.  Uncertainties can only be assessed against independent observations and so far independent measurements of mass loading are extremely sparse, let alone independent measurements of atmospheric ash particle size distributions, shapes and composition."

I have also made this change: "In the case of the ash retrievals for Grímsvötn, the error estimates are within the expected range, giving an error of ±0.1 Tg or roughly 20--50% of the estimated mass of very fine ash.  It is emphasized that this is not the total mass emitted by the volcano, which is typically a few percent of the total mass.  It is however, the mass fraction that is dispersed by the winds and the very fine ash that can cause damage to aircraft jet engines."

I have also fitted in a reference to Stevenson et al. (2015) but as indicated above this is contentious and any detailed discussion of that work requires a more considered approach.  Also I don't understand why I should need to discuss someone else's work in my paper – it is now referenced.   After (Section 4.5), I have added: "The error (precision) in estimating very fine ash mass ..."

"Stevenson et al. (2015) discuss potential errors in satellite retrievals by using cryptotephra data to speculate that larger particles exist in dispersing ash clouds (although no atmospheric observations are presented) and claim through modelling studies that current retrieval schemes (all of them) underestimate mass loadings because of the dense sphere assumption and lack of sensitivity to particles with diameters > 10 μm."

Figure 4.  Caption.  Added: "Isolines (contours) of brightness temperatures are shown in white to highlight the location and expansion of the top of the column."

Figure 7. Font size increased.